# Unsupervised Discovery of Object Radiance Fields

**Hong-Xing Yu**
Stanford University

**Leonidas J. Guibas**
Stanford University

**Jiajun Wu**
Stanford University

## ABSTRACT

We study the problem of inferring an object-centric scene representation from a single image, aiming to derive a representation that is learned without supervision, explains the image formation process, and captures the scene's 3D nature. Most existing methods on scene decomposition lack one or more of these characteristics, due to the fundamental challenge in integrating powerful unsupervised inference schemes like deep networks with the complex 3D-to-2D image formation process. In this paper, we propose unsupervised discovery of Object Radiance Fields (uORF), integrating recent progresses in neural 3D scene representations and rendering with deep inference networks for unsupervised 3D scene decomposition. Trained on only multi-view RGB images, uORF learns to decompose complex scenes with diverse, textured background from a single image. We show that uORF enables novel tasks, such as scene segmentation and editing in 3D, and it performs well on these tasks and on novel view synthesis on three datasets[*].

## 1 INTRODUCTION

Building factorized, object-centric scene representations is a fundamental ability in human vision and a constant topic of interest in computer vision and machine learning. We identify that such representations should bear three characteristics: they should be learned without supervision or prior knowledge about object categories, and therefore applicable to environments where object categories are unknown; they should explain the image formation process, addressing questions like 'what if the object is not there?'; they should be 3D-aware, capturing geometric and physical object properties for navigation, interaction, and manipulation.

For decades, researchers have attempted to solve the problems from various angles. Inspiring as they are, these methods each lack in one or more of the three aspects (Table 1). Computer vision research on unsupervised object discovery has achieved great success on deriving object segments from real images, but it doesn't capture the image formation process, nor is it 3D-aware (Rubinstein et al., 2013; Zhu et al., 2012). Recent work on deep probabilistic inference for visual scene decomposition is unsupervised and generative (Burgess et al., 2019; Engelcke et al., 2019; Locatello et al., 2020), though most still formulate the problem as 2D segmentation and work on simple scenes of geometric primitives, ignoring the complex 3D nature of realistic visual scenes. A few recent papers on 'scene de-rendering' have attempted to reconstruct 3D, object-centric representations by leveraging the forward rendering procedure (Yao et al., 2018; Ost et al., 2021); they are however supervised, relying on annotations of specific object and scene categories, such as cars and road scenes.

The fundamental challenge that prevents these systems from acquiring all three desired properties is that the image formation process from 3D to 2D is complex and non-differentiable (e.g., due to occlusion). Thus, for a long time, it has been unclear how it may be integrated with powerful deep inference schemes. But most recently, progresses in neural rendering (Tewari et al., 2020) have demonstrated that their continuous, implicit representation works well with gradient-based inference models, such as deep networks. In particular, Neural Radiance Fields (NeRFs) (Mildenhall et al., 2020) recover a 3D scene from a set of RGB images via differentiable volume rendering. Such encouraging advances in generative modeling suggest a promising route for inferring 3D, generative, and object-centric scene representations without supervision.

In this paper, we propose unsupervised discovery of Object Radiance Fields (uORF), integrating conditional NeRFs as 3D object representations with deep inference networks for unsupervised 3D

---

[*]Code and data can be found at `https://kovenyu.com/uORF/`.

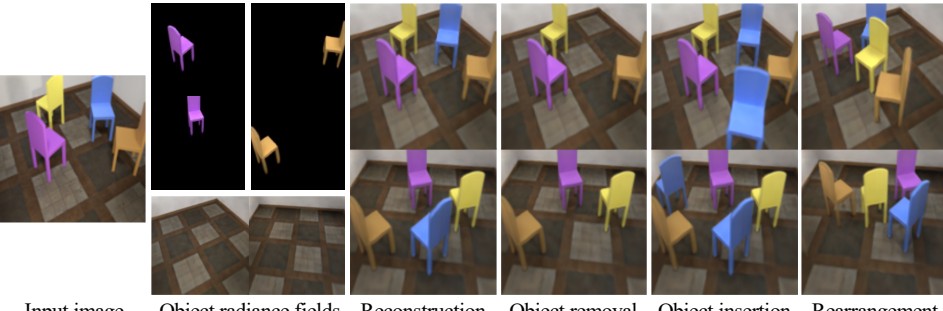

Input image    Object radiance fields    Reconstruction    Object removal    Object insertion    Rearrangement
Top: objects; bottom: background      Top: input view; bottom: novel view

Figure 1: Illustration of unsupervised discovery of Object Radiance Fields. We aim to infer factorized object and background radiance fields from a single view, allowing reconstructing and editing of the scene.

scene decomposition. uORF infers a set of object radiance fields and a background radiance field; thus, uORF represents a 3D scene as a composition of object radiance fields (Figure 1). During training, such radiance fields are neurally rendered in multiple views, with reconstruction losses in pixel space as training supervision; during testing, uORF infers the set of object radiance fields from a single image. Learning uORF does not require explicit supervision of 3D geometry or object segmentation, but only sparse multi-view RGB images of training scenes.

The integration of NeRFs allows us to work with more realistic scenes with complex object shapes and diverse background environments, beyond simple scenes such as those in multi-dSprites (Greff et al., 2019) and CLEVR (Johnson et al., 2017), as considered by most current unsupervised scene decomposition methods. We further make two innovations to improve uORF's performance. First, as background geometry and appearance can be quite different from foreground objects in 3D, we design uORF with explicit modeling of both components. This background-aware design not only facilitates learning on complex scenes, but also allows single-image scene editing including moving individual objects and changing background. Second, as volume rendering requires massive queries to render a single pixel for the recomposed scene, a practical challenge of learning uORF lies in the computational inefficiency. We tackle this issue by proposing a novel progressive coarse-to-fine training which improves representation quality while remaining affordable computational cost.

We evaluate uORF on factorized scene representation learning (e.g., segmentation in 3D) and scene generation (e.g., novel view synthesis, scene editing in 3D). Our evaluation is on three datasets with a gradually increasing complexity: first, CLEVR-like scenes with primitives foreground shapes; second, room scenes with complex chair shapes and textured backgrounds; third, more diverse room scenes with various foreground shapes and backgrounds. Our results show that uORF learns factorized representations that can segment 3D scenes into objects with fine shape details (e.g., thin chair legs) and backgrounds with well-recovered appearance details (e.g., irregular textures of a wooden floor).

In summary, our contributions are three-fold. First, we propose the problem of inferring an unsupervised, factorized, generative, and 3D-aware scene representation from a single image. Second, we introduce unsupervised discovery of Object Radiance Fields (uORF) that infers individual 3D object radiance fields from a single view for the proposed problem. Third, we demonstrate that uORF enables novel tasks such as scene segmentation and editing in 3D, and we show that it generalizes to novel scene arrangement and unseen combinations of object properties.

## 2   RELATED WORK

**Co-segmentation and object discovery.** Our work is closely related to traditional computer vision methods on object discovery, which aims to locate (visually similar) objects in a collection of images. These methods typically model objects as visual words and adopted methods from topic modeling to localize objects (Russell et al., 2006; Sivic et al., 2005; 2008), or cluster and group image patches (Grauman & Darrell, 2006; Joulin et al., 2010; Rubio et al., 2012; Vicente et al., 2011; Rubinstein et al., 2013; Cho et al., 2015). Recent works have integrated the clustering-based strategy with deep learning (Li et al., 2019; Vo et al., 2020). Nevertheless, they do not explain image formation process nor are they 3D-aware.

| Approach | Unsup. | Gen. | 3D |
|---|---|---|---|
| Co-segmentation | ✓ | ✗ | ✗ |
| Deep prob. infer. | ✓ | ✓ | ✗ |
| Scene "de-render" | ✗ | ✓ | ✓ |
| Ours | ✓ | ✓ | ✓ |

Table 1: Comparison to existing methods.

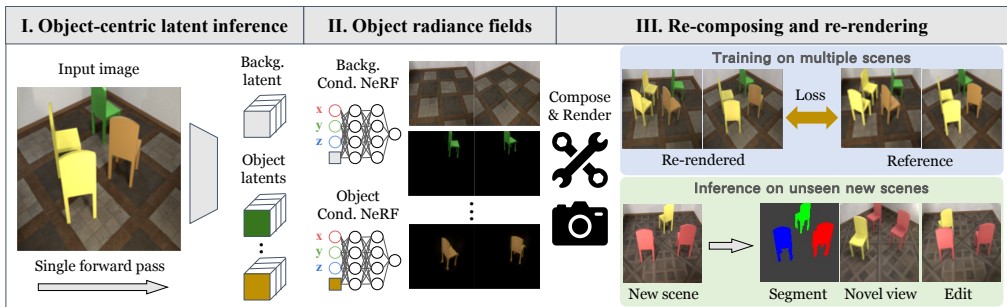

Figure 2: **Overview**. **I.** Our model learns to infer a set of latents in a single forward pass. **II.** Each object/background radiance field consists of a latent and a shared conditional NeRF. **III.** During training, we recompose the scene and re-render images for supervision. We train our model on different scenes. At test time, we use a single image of an unseen scene for reconstruction or editing.

**Deep probabilistic inference for scene decomposition.** Our method is also closely related to deep probabilistic inference for scene decomposition. Most works formulate the problem as compositional generative modeling, where a visual scene is represented by a set of latent codes that either correspond to localized object-centric patches (Eslami et al., 2016; Crawford & Pineau, 2019; Kosiorek et al., 2018; Lin et al., 2020; Jiang et al., 2019) or scene mixture components (Burgess et al., 2019; Greff et al., 2019; 2016; 2017; Engelcke et al., 2019). Recently, Locatello et al. (2020) proposed the Slot Attention module to simplify the inference by a slot-based encoder. Besides these inference models, Monnier et al. (2021) formulated scene decomposition as layered image decomposition and demonstrated it on real images. However, these methods do not account for the 3D nature of scenes.

A few methods have recently been proposed for unsupervised 3D scene decomposition. Elich et al. (2020) infer object shapes from a single scene image, but they require pretraining on groundtruth shapes. Chen et al. (2020) extend Generative Query Network (Eslami et al., 2018) to decompose 3D scenes, but they require multi-view images during inference. The closest to our work is a concurrent work by Stelzner et al. (2021) which also utilizes a slot-based encoder and NeRFs as 3D representations. However, Stelzner et al. (2021) relies on groundtruth multi-view dense depth in addition to images in training. Moreover, we explicitly model the separation of objects and background to address various complex shapes and textured backgrounds, while they only demonstrate scenes with a single textureless background.

**Scene de-rendering.** A few recent works have shown reconstructing 3D object-centric representations by incorporating forward image rendering process (Wu et al., 2017; Yao et al., 2018; Kundu et al., 2018; Ost et al., 2021). Yao et al. (2018) de-render an image into semantic segments and geometric object attributes, which enable 3D scene manipulation. Most recently, Ost et al. (2021) propose Neural Scene Graph to represent dynamic scenes into a scene graph where each node encodes object-centric information. However, these methods rely on manual annotations of specific objects (such as cars) and scene categories (such as street scenes).

**Neural scene representations and rendering.** Our method is related to recent progresses in neural continuous scene representations (Sitzmann et al., 2019) and neural rendering (Tewari et al., 2020). Neural scene representations parameterize 3D scenes with a deep network (Sitzmann et al., 2019). Combined with differentiable neural rendering techniques (Kato et al., 2020; Tewari et al., 2020), they can be learned from only 2D images (Niemeyer et al., 2020). In particular, Neural Radiance Fields (NeRFs) (Mildenhall et al., 2020) have shown impressive novel view synthesis. Related follow-up works include those that infer NeRFs from a single image (Yu et al., 2020; Kosiorek et al., 2021; Jang & Agapito, 2021) and those that incorporate NeRFs into generative models (Schwarz et al., 2020; Niemeyer & Geiger, 2021; Chan et al., 2020). Different from these works which cope with single objects or holistic scenes, we learn object NeRFs via decomposing a multi-object scene without segmentation annotations. GIRAFFE (Niemeyer & Geiger, 2020) generates object NeRFs and thus compose 3D scenes in an adversarial framework. However, it targets at unconditional generation and cannot tackle inference (see Appendix E), while we focus on single-image inference of multi-object scenes. Thus, we address a fundamentally different problem compared to GIRAFFE.

## 3 APPROACH

Our goal is to infer from a single image a set of object-centric 3D representations to generate the underlying 3D scene. We show an illustration in Figure 2. Our object representation is a conditional

object radiance field. Thus, we learn to infer object-centric latents from a single image (Figure 2-I). The inferred latents are used to condition a network to yield the 3D object and background radiance fields (Figure 2-II), forming our 3D-aware, generative and factorized scene representation. In training, we compose all object and background radiance fields and render the recomposed scene from multiple views. We obtain supervision by comparing rendered images to reference images (Figure 2-III) without needing 3D geometry or segmentation annotations. We describe each of our model components in the following and leave implementation details in Appendix B.

## 3.1 OBJECT-CENTRIC LATENT INFERENCE

Our goal is to infer latent object-centric representations from a single input image. We assume that an underlying 3D scene is composed of a background environment and no more than $K$ foreground objects. Thus, the output of our object-centric latent inference process is a latent $\mathbf{z}^b$ for background and a set of latents $\{\mathbf{z}_i^f\}_{i=1}^K$ for foreground objects (empty objects are allowed). To encourage unsupervised object-wise factorization, we adopt a slot-based formulation (Locatello et al., 2020). The assumption in this formulation is that objects should share a common prior latent space. The main idea include three steps. The first step is to sample all object latents (i.e., slots) from the same prior

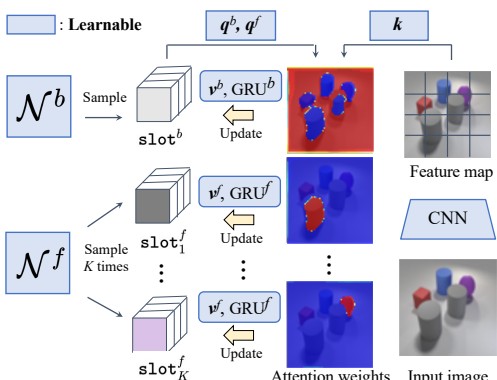

Figure 3: Our object-centric latent inference. The attention binds each object's features to a slot.

distribution (background is a special object) to encourage representational uniformity across all slots ("*sampling*"). Then each slot is bound to an object region via an attention module ("*binding*"). In the last step each slot gets updated by the bound object features to specialize for that object ("*updating*"). Locatello et al. (2020) have demonstrated success on segmenting 2D images.

However, in 3D scenes, the geometry and appearance of the background are highly different from those of foreground objects. Modeling them indistinguishably often leads to object representations entangled with blurry background segments (Burgess et al., 2019; Locatello et al., 2020), which impedes applications such as scene editing and re-composition. Thus, we propose a background-aware slot attention module (Figure 3) that separately models objects and environment to better capture the compositional structure of 3D scenes.

**Background-aware slot attention for sampling and binding.** In the sampling step, we model the latent prior distribution of foreground objects by a Gaussian with learnable mean and variance, i.e., we sample $\texttt{slots}^f \sim \mathcal{N}^f(\mu^f, \text{diag}(\sigma^f)) \in \mathbb{R}^{K \times D}$ for $K$ objects. For latent prior of backgrounds, we learn another Gaussian and sample a single slot from it, i.e., $\texttt{slot}^b \sim \mathcal{N}^b(\mu^b, \text{diag}(\sigma^b)) \in \mathbb{R}^{1 \times D}$.

To bind slots to image features, we let all the slots to compete for explaining the input image representation. To do this, we flatten the convolutional feature map (we include details about convolutional encoder in Appendix B.1) into a set of $N$ input feature vectors, $\texttt{feat} \in \mathbb{R}^{N \times D}$. The slot competition is modeled by a key-query attention (Bahdanau et al., 2014):

$$\texttt{attn}_{i,j} := \frac{\exp(M_{i,j})}{\sum_l \exp(M_{i,l})}, \quad \text{where} \quad M := \frac{1}{\sqrt{D}} k(\texttt{feat}) \cdot \begin{bmatrix} q^b(\texttt{slot}^b) \\ q^f(\texttt{slots}^f) \end{bmatrix}^T \in \mathbb{R}^{N \times (K+1)}. \quad (1)$$

Here $k$ and $q^b/q^f$ are learnable linear mappings $\mathbb{R}^{D \to D}$ for computing dot-product similarity (Luong et al., 2015), and $\sqrt{D}$ is a fixed softmax temperature (Vaswani et al., 2017). One can see this process as a soft K-means, where $\texttt{attn}_i$ softly assigns a feature $i$ to the slots (centroids). The background slot is expected to capture the modality of background features and bind all of them, allowing foreground slots to focus only on the objects without explaining background segments (Figure 3). Besides the representation design, we further encourage disentanglement between background and foregrounds by two additional designs: (1) We represent and query foreground/background (during the neural rendering process) in different coordinate frames. (2) To discourage object slots from fitting background, we impose a locality constraint in early training. We set a foreground box and enforce that every foreground query point outside the box has zero density. We include details in Appendix B.2.

**Updating slots to infer latents.** With the attention weights, we form the update signal by aggregating input values via a weighted mean pooling $\texttt{updates}^b := W^{bT} \cdot v^b(\texttt{feat}) \in \mathbb{R}^{1 \times D}$, where $W_{i,1}^b := \texttt{attn}_{i,1}/(\sum_{l=1}^N \texttt{attn}_{l,1})$, and $\texttt{updates}^f := W^{fT} \cdot v^f(\texttt{feat}) \in \mathbb{R}^{K \times D}$, where $W_{i,j}^f := \texttt{attn}_{i,j+1}/(\sum_{l=1}^N \texttt{attn}_{l,j+1})$. Slots are then updated using the update signals via a learnable rule parameterized by a Gated Recurrent Unit (GRU) (Cho et al., 2014), so that $\texttt{slots}^f \leftarrow \texttt{GRU}^f(\texttt{slots}^f, \texttt{updates}^f)$ and $\texttt{slot}^b \leftarrow \texttt{GRU}^b(\texttt{slot}^b, \texttt{updates}^b)$. We repeat the attention computation and updating for 3 iterations, and output all the slots as the final latents $\mathbf{z}^b$ and $\{\mathbf{z}_i^f\}_{i=1}^K$. We show pseudo-code of our background-aware slot attention in Appendix (Alg. 1).

## 3.2 COMPOSITIONAL NEURAL RENDERING

We represent a 3D object as a conditional neural radiance field. A NeRF is a continuous mapping $g : (\mathbf{x}, \mathbf{d}) \to (\mathbf{c}, \sigma)$ from spatial location $\mathbf{x}$ and viewing direction $\mathbf{d}$ to emitted color $\mathbf{c}$ and volume density $\sigma$ used for volume rendering (Max, 1995). This mapping is parameterized by an MLP network. We adopt a conditional NeRF $g(\mathbf{x}, \mathbf{d}|\mathbf{z})$ for our inference scheme (detailed in Appendix B.3). The MLP parameters are shared across all objects $g^f(\mathbf{x}, \mathbf{d}|\mathbf{z}_i^f)$, but not the background $g^b(\mathbf{x}, \mathbf{d}|\mathbf{z}^b)$ due to its distinct geometry and appearance distribution.

To compose individual objects and background into the holistic scene, we consider a scene mixture model and use density-weighted mean to combine all components: $\bar{\sigma} = \sum_{i=0}^K w_i \sigma_i, \bar{\mathbf{c}} = \sum_{i=0}^K w_i \mathbf{c}_i$, where $w_i = \sigma_i/\sum_{j=0}^K \sigma_j$. Here $\bar{\sigma}$ and $\bar{\mathbf{c}}$ are the combined density and color, respectively. The color $C(\mathbf{r})$ of a camera ray $\mathbf{r}(t) = \mathbf{o} + \mathbf{d}(t)$ is then estimated via numerical integration of volume rendering, using $S$ discrete combined samples along a ray (Max, 1995): $C(\mathbf{r}) = \sum_{i=1}^S T_i[1 - \exp(-\bar{\sigma}_i \delta_i)]\bar{\mathbf{c}}_i$, where $T_i = \exp\left(-\sum_{j=1}^{i-1} \bar{\sigma}_j \delta_j\right)$. Here $\delta_j$ is the distance between adjacent samples along a ray.

## 3.3 MODEL LEARNING

**Loss functions.** As shown in Figure 2, during training we input a single image of a scene, infer object and background radiance fields, render multiple views from the recomposed scene, and compare them to reference images for loss computation. We train our model across multiple scenes. Our training loss function comprises of a reconstruction loss, a perceptual loss, and an adversarial loss: $\mathcal{L} = \mathcal{L}_{\text{recon}} + \lambda_{\text{percept}}\mathcal{L}_{\text{percept}} - \lambda_{\text{adv}}\mathcal{L}_{\text{adv}}$, where $\lambda$ are weights. The reconstruction loss is $\mathcal{L}_{\text{recon}} = \|\boldsymbol{I} - \hat{\boldsymbol{I}}\|^2$, where $\boldsymbol{I}$ and $\hat{\boldsymbol{I}}$ denote the reference image and rendered image, respectively.

Since we estimate 3D radiance fields from a single view, there can be uncertainties about the appearance from other views (e.g., the back view). For example, regarding visual appearance of objects, inaccurate global lighting estimation leads to uncertainties in brightness and shadows from occluded views even if the object shapes can be well estimated. To address this, we incorporate a perceptual loss (Johnson et al., 2016) which is tolerant to mild appearance changes. The perceptual loss is defined by $\|\mathcal{L}_{\text{percept}} = p(\boldsymbol{I}) - p(\hat{\boldsymbol{I}})\|^2$ where $p$ is a deep feature extractor (See Appendix B.4).

In addition to appearance, there can be even higher uncertainties in estimating object shapes from a single view, which is a multi-modal distribution. In this case, the unimodal reconstruction loss leads to blurry results ("mean shape"). We mitigate this issue by adding an adversarial loss which can deal with multi-modal distributions:

$$\mathcal{L}_{\text{adv}} = \mathbb{E}[f(D(\hat{\boldsymbol{I}}))] + \mathbb{E}[f(-D(\boldsymbol{I})) + \lambda_R\|\nabla D(\boldsymbol{I})\|^2], \quad \text{where} \quad f(t) = -\log(1 + \exp(-t)). \quad (2)$$

Here we adopt the R1 regularization (Mescheder et al., 2018) to stabilize training. $D$ denotes a discriminator to distinguish rendered images $\hat{\boldsymbol{I}}$ and reference images $\boldsymbol{I}$. We iterate between training the discriminator by minimizing $\mathcal{L}_{\text{adv}}$ and training our inference model (Figure 2) by minimizing $\mathcal{L}$.

**Coarse-to-fine Progressive Training.** A practical challenge in training compositional NeRFs lies in the computational cost of neural volume rendering, as it requires massive queries to render a single pixel. While there have been attempts on fast inference (Liu et al., 2020; Rebain et al., 2020; Neff et al., 2021; Garbin et al., 2021; Reiser et al., 2021; Yu et al., 2021), high space complexity in training remains a challenge. Further, because our perceptual and adversarial losses depend on image patches, the system has to render a large enough patch (instead of a single pixel) at the same time, which further increases its space demand.

To allow training on a higher resolution, we propose a coarse-to-fine progressive training. In a coarse training stage, we bilinearly downsample reference images to a low resolution (e.g., $64 \times 64$), and train

uORF on these downsampled images. Although the coarsely trained model can already decompose the 3D scenes and recover rough object radiance fields, fine details (e.g., thin legs of chairs) might be missing. Thus, in a following fine training stage, we replace the low-resolution reference images with image patches randomly cropped from high-resolution images (Figure 2-III), and render the correspondingly located patches from our recomposed scene radiance fields to compute the loss. We include further training details in Appendix B.4.

## 4 EXPERIMENTS

We evaluate uORF on both scene representation (via scene segmentation in 3D) and scene generation (via novel view synthesis and scene editing) on three datasets.

**Data.**    We build three synthetic datasets with gradually increasing complexity. For each scene in the dataset, we point the camera to the scene center and render four images with a randomly chosen azimuth angle and a fixed elevation angle. We describe more details in Appendix C.1.

CLEVR-567. The first dataset includes scenes of 5–7 CLEVR objects (Johnson et al., 2017), with a random position and orientation on a clean background. Foreground object shapes include three geometric primitives (i.e., cubes, spheres and cylinders). Since there is intrinsic ambiguity in estimating specularity from a single image, we use only the largely diffuse "Rubber" material. There are 1,000 scenes for training and 500 for testing.

Room-Chair. The second dataset includes scenes of 3 to 4 chairs of the same shape in a room with three different textured backgrounds. There are 1,000 scenes for training and 500 for testing.

Room-Diverse. The third dataset includes scenes of diverse foreground object shapes and background appearances. Each scene includes 4 different chairs, whose shape is randomly sampled from 1,200 ShapeNet chair shapes (Chang et al., 2015), and the background is sampled from 50 floor textures from the web. There are 5,000 scenes for training and 500 for testing.

### 4.1 SCENE SEGMENTATION IN 3D

We first evaluate uORF's factorized 3D scene representations via scene segmentation in 3D.

**Baselines.**    Because there is no previous work focusing on the same setting as uORF, we compare to a 2D state-of-the-art scene decomposition model Slot Attention (Locatello et al., 2020) for unsupervised scene segmentation wherever possible (detailed in Appendix C.2). In addition, we compare to two ablated versions of uORF. First, we remove our background-aware modeling but keep the same number of slots. Second, we ablate our progressive training such that the training procedure only contains the coarse training stage. We refer to ablated models as "uORF (w/o background)" and "uORF (w/o prog. train.)", respectively.

**Metrics.**    We adopt the widely-used Adjusted Rand Index (ARI) as our metric. To evaluate scene segmentation in 3D, we consider three kinds of ARIs: (1) For direct comparison to 2D methods, we compute ARI on reconstructed images. (2) To reflect the 3D nature, we also compute ARI on synthesized novel views, denoted as "NV-ARI". Note that each scene includes 4 views, and only one is used as input, and the other three are treated as novel views for this metric. (3) In line with previous 2D methods, we also report foreground ARI (Fg-ARI), computed only on foreground regions indicated by groundtruth masks. Yet, we note that Fg-ARI cannot fully reflect the segmentation quality, because background segmentation is completely ignored.

**Results.**    We volume-render a density map $\mathbf{d}^i$ for each slot $i$. The segmentation label for each pixel $s_p$ is given by $s_p = \arg\max_{i=1}^{K+1} \mathbf{d}_p^i$. We show results on Table 2 and Figure 4 (more in Appendix D). For all segmentation metrics, we show mean and standard deviation for three runs. uORF outperforms all methods in terms of ARI and NV-ARI. From Figure 4, it is clear that uORF is able to discover the 3D objects from a single image. These results validate that uORF can learn well-factorized 3D object-centric scene representations. Also notice that uORF yields better ARI even in input views compared to 2D slot attention. This is likely due to our background-aware design, as our ablated model "uORF w/o background" has shown similar input-view results compared to slot attention (e.g., see 3rd and 4th columns in Figure 4).

### 4.2 NOVEL VIEW SYNTHESIS

We then show that uORF is 3D-aware and generative via evaluation on novel view synthesis.

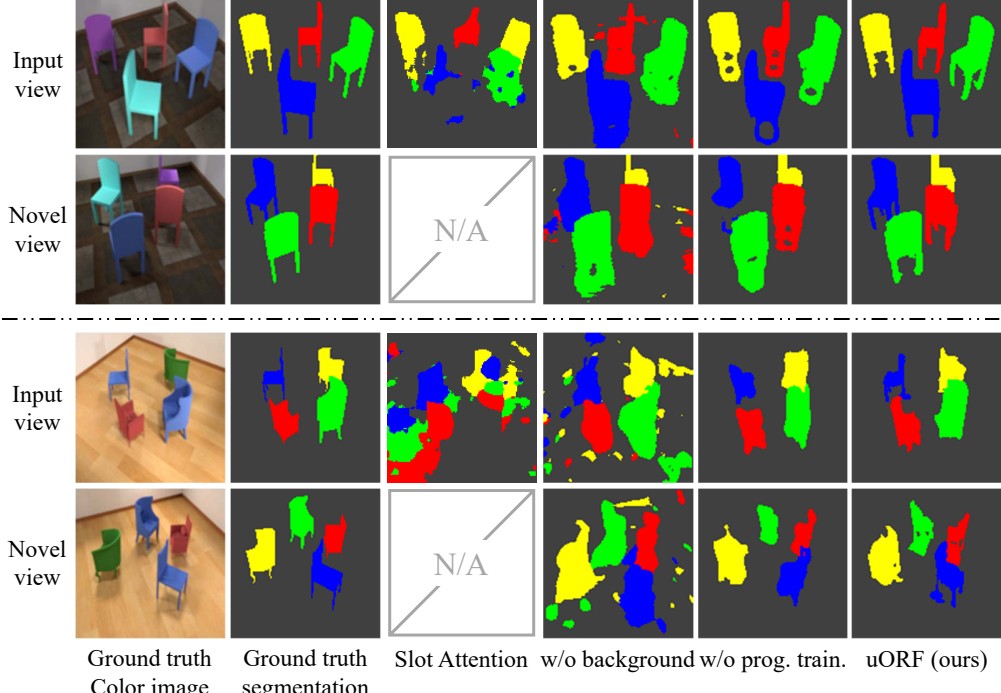

Figure 4: Examples on scene segmentation in 3D. Novel view images are for reference but not input.

| Models | CLEVR-567 | | | Room-Chair | | | Room-Diverse | | |
|---|---|---|---|---|---|---|---|---|---|
| | 3D metric | 2D metric | | 3D metric | 2D metric | | 3D metric | 2D metric | |
| | NV-ARI↑ | ARI↑ | Fg-ARI↑ | NV-ARI↑ | ARI↑ | Fg-ARI↑ | NV-ARI↑ | ARI↑ | Fg-ARI↑ |
| Slot Attention | N/A | 3.5±0.7 | **93.2**±1.5 | N/A | 38.4±18.4 | 40.2±4.5 | N/A | 17.4±11.3 | 43.8±11.7 |
| uORF (w/o background) | 10.5±3.6 | 11.7±4.6 | 86.4±2.8 | 40.4±9.2 | 42.3±10.6 | **93.3**±1.9 | 21.0±8.1 | 24.0±9.9 | **78.9**±3.1 |
| uORF (w/o prog. train.) | 81.1±0.7 | 83.7±0.8 | 84.2±0.5 | 62.3±2.5 | 65.4±2.6 | 81.0±3.0 | 53.8±1.4 | 63.7±1.7 | 66.9±4.1 |
| uORF (ours) | **83.8**±0.3 | **86.3**±0.1 | 87.4±0.8 | **74.3**±1.9 | **78.8**±2.6 | 88.8±2.7 | **56.9**±0.2 | **65.6**±1.0 | 67.9±1.7 |

Table 2: Scene segmentation results. "NV-ARI" refers to ARI evaluated on novel views. "Fg-ARI" refers to ARI evaluated with only foreground pixels. Slot Attention (Locatello et al., 2020) is a state-of-the-art 2D method.

**Setup.** For each test scene, we randomly pick one image as input and the remaining three images as groundtruth for novel view synthesis. As Slot Attention is purely in 2D and does not support novel view synthesis, we compare to a conditional NeRF (Mildenhall et al., 2020), equipped with a convolutional encoder similar to uORF, termed as "NeRF-AE" (see Appendix C.2). For fair comparison, we increase the latent dimension for NeRF-AE to guarantee approximately the same computational cost, and we use the same training strategy and losses as uORF. Thus, NeRF-AE can also be seen as a monolithic alternative model to uORF. We also compare with the ablated models, "uORF (w/o background)" and "uORF (w/o prog. train.)". We use the perceptual metric LPIPS (Zhang et al., 2018), together with SSIM (Wang et al., 2004) and PSNR, as our evaluation metrics.

**Results.** Quantitative results are in Table 3 and qualitative results are in Figure 5 (more in Appendix D). Quantitatively, uORF outperforms all compared methods on all metrics. From the qualitative comparison in Figure 5, we highlight three advantages of uORF. First, compared with NeRF-AE, which has a monolithic latent structure for the entire scene, uORF better preserves the features of each object: for example, see how NeRF-AE fuses object colors in the first two rows, while uORF does not. This shows the advantage of factorized scene representations to structurally describe a visual scene. Second, compared with uORF (w/o background), one can clearly see how our background-aware modeling helps recovering background appearances: uORF can accurately recover background appearance of the Room-Chair example, while uORF (w/o background) does not. It also facilitates learning on complex scenes with diverse, textured background: uORF can learn to roughly recover object shapes in the Room-Diverse example. Third, compared with uORF (w/o prog. train.), we highlight that the fine training on image patches indeed improves both visual quality and representation quality: the full uORF tries to recover sharp edges of cubes, while uORF (w/o prog. train.) cannot distinguish cube from sphere.

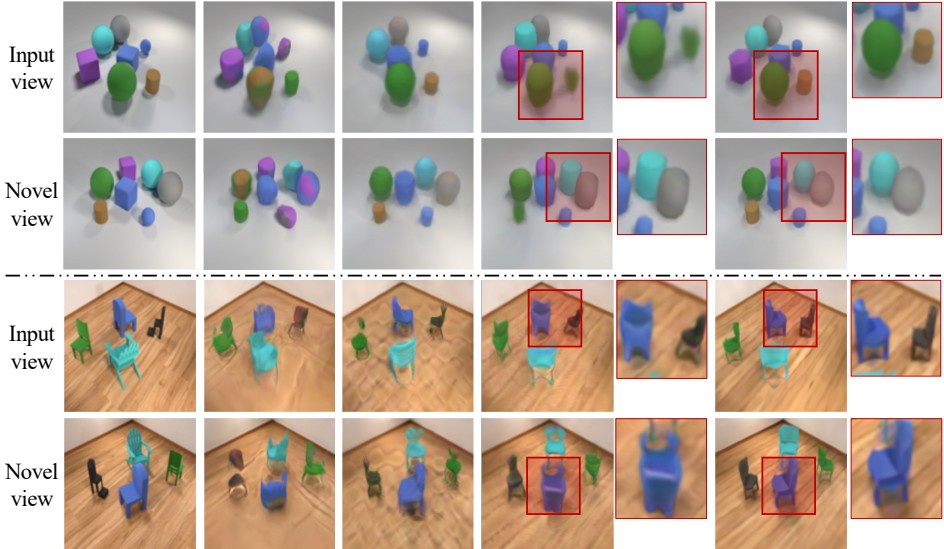

Figure 5: Qualitative results on scene decomposition and novel view synthesis. Within every two rows, the first is reconstruction and the second is a novel view.

| Models | CLEVR-567 | | | Room-Chair | | | Room-Diverse | | |
|---|---|---|---|---|---|---|---|---|---|
| | LPIPS↓ | SSIM↑ | PSNR↑ | LPIPS↓ | SSIM↑ | PSNR↑ | LPIPS↓ | SSIM↑ | PSNR↑ |
| NeRF-AE | 0.1288 | 0.8658 | 27.16 | 0.1166 | 0.8265 | 28.13 | 0.2458 | 0.6688 | 24.80 |
| uORF (w/o background) | 0.0919 | 0.8924 | 28.93 | 0.1671 | 0.7852 | 27.86 | 0.2231 | 0.6924 | 25.90 |
| uORF (w/o prog. train.) | 0.1044 | 0.8894 | 28.84 | 0.1573 | 0.8287 | 28.33 | 0.2123 | 0.6760 | 25.19 |
| uORF (ours) | **0.0859** | **0.8971** | **29.28** | **0.0821** | **0.8722** | **29.60** | **0.1729** | **0.7094** | **25.96** |

Table 3: Comparison on novel view synthesis from a single image.

Overall, the novel view synthesis results suggest that uORF can learn to represent 3D scenes with reasonable fidelity, even with the presence of complex foreground object shapes, such as chairs and different textured backgrounds.

### 4.3 SCENE DESIGN AND EDITING IN 3D

Being object-centric and 3D-aware, uORF is able to edit 3D scene radiance fields inferred from a single view, and generate novel scene images.

**Setup.** We test uORF's ability to edit scenes and synthesize novel images on the Room-Chair dataset. We consider both moving foreground objects and changing background appearance. For object moving, we randomly pick one object in a test scene and move it to a random position. We render 4 images for each of the 500 test scenes. For background changing, we replace the current background texture to a different one and also render 4 images for evaluation. To indicate the new background, we re-pick and re-put foreground objects such that the resultant background indicator image is different from the groundtruth image.

For uORF and Slot Attention (Locatello et al., 2020), we use groundtruth masks of the input view only for ease of evaluation. We determine which slot to move by picking the one with largest mask IoU. For NeRF-AE (Mildenhall et al., 2020) to do editing, we back-project the masks to frustums to determine the 3D regions to be moved/replaced. We use LPIPS, SSIM, and PSNR as our metrics.

**Results.** We show results in Table 4 and Figure 6 (more in Appendix D). Again, uORF outperforms all compared methods on all metrics. As Figure 6 depicts, images synthesized by uORF show least artifacts and highest quality and fidelity.

| Models | Moving objects | | | Changing background | | |
|---|---|---|---|---|---|---|
| | LPIPS↓ | SSIM↑ | PSNR↑ | LPIPS↓ | SSIM↑ | PSNR↑ |
| NeRF-AE | 0.2451 | 0.7284 | 23.18 | 0.2185 | 0.7132 | 25.42 |
| Slot Attention | 0.3941 | 0.7134 | 23.06 | 0.3689 | 0.7283 | 23.94 |
| uORF (w/o background) | 0.2206 | 0.7448 | 24.55 | 0.1879 | 0.7719 | 26.68 |
| uORF (w/o prog. train.) | 0.1583 | 0.8313 | 28.19 | 0.1586 | 0.8306 | 28.27 |
| uORF (ours) | **0.0855** | **0.8711** | **29.26** | **0.0822** | **0.8729** | **29.53** |

Table 4: Comparison on scene editing.

### 4.4 GENERALIZATION AND ANALYSIS

Finally we explore the generalization ability of uORF. We consider generalization on unseen, challenging spatial arrangement of objects, as well as generalization on unseen object appearances.

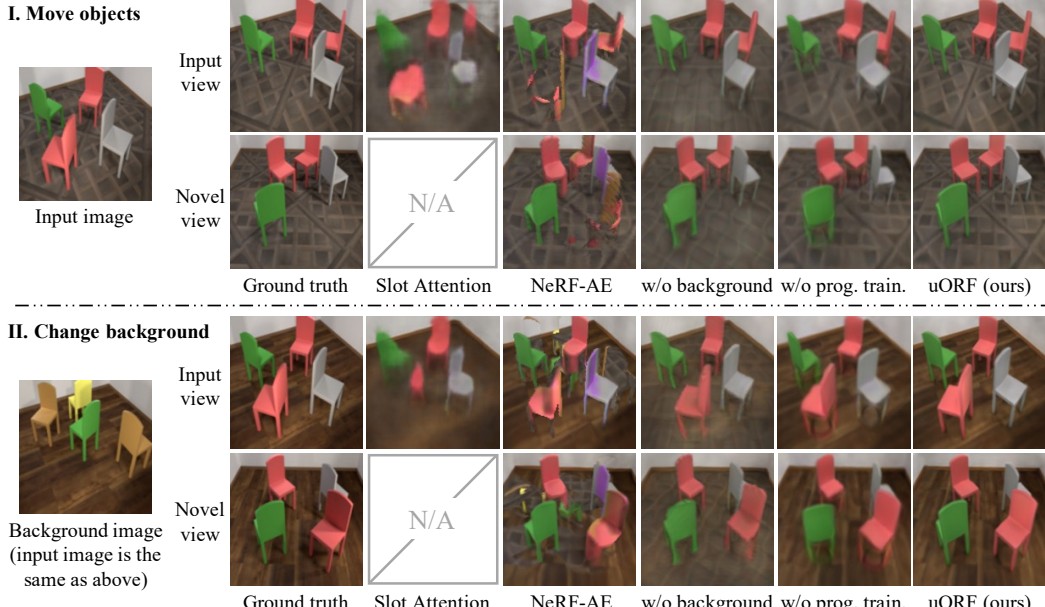

Figure 6: Qualitative results on single-image 3D scene manipulation. The first two rows are for moving object and the second two rows are for changing background.

| Models | ARI↑ | LPIPS↓ |
|---|---|---|
| Slot Attention | 5.7±0.3 | N/A |
| NeRF-AE | N/A | 0.2201 |
| uORF (ours) | **83.2**±0.6 | **0.1540** |

Table 5: Generalization to novel challenging spatial arrangements.

| Models | NV-ARI↑ | ARI↑ |
|---|---|---|
| Slot Attention | N/A | 2.2±0.6 |
| uORF (ours) | 85.0±0.3 | 87.4±0.4 |
| uORF (oracle) | **85.5**±0.3 | **87.5**±0.3 |

Table 6: Generalization to unseen combinations of color and shape.

| Loss functions | ARI↑ | LPIPS↓ |
|---|---|---|
| Rec. | 59.1±0.5 | 0.3610 |
| Rec. + Percept. | 65.2±0.8 | 0.2156 |
| Rec. + Adv. | 60.4±2.2 | 0.2288 |
| Rec. + Percept. + Adv. | **65.6**±1.0 | **0.1729** |

Table 7: Ablation study for losses on the Room-Diverse dataset.

**Generalizing to challenging spatial arrangements.** We build a new test dataset, packed-CLEVR-11, where each scene has 11 objects that are closely packed into a cluster. Therefore, each scene bears an unseen number of objects in an unseen challenging arrangement. We test models trained on CLEVR-567, report results in Table 5 and Appendix Figure 19. Despite uORF never sees such object arrangements, it still achieves a reasonable performance and outperforms baselines.

**Generalizing to new combination of shape and color.** For unseen object appearances, we consider generalization in a systematic way such that the model can deal with unseen combination of object color and shape. Thus, we build a new training set similar to CLEVR-567, but we remove red cylinders and blue spheres from the object candidate pool. Then we test trained models on another dataset with only red cylinders and blue spheres in the candidate pool. We show results in Table 6 and examples in Appendix Figure 20. We see that although uORF has never seen any of the test set objects, it achieves similar results to the one trained on a normal CLEVR-567 dataset (denoted as "uORF (oracle)"). This suggests uORF's ability for systematic generalization to unseen combinations of object color and shape. We further validate generalization to unseen object shapes in Appendix D.

**Evaluating loss functions.** uORF uses perceptual and adversarial losses to combat intrinsic uncertainties in single-image inference of 3D representations. We show ablation results on novel view synthesis in Table 7 and Appendix Figure 18. Both losses significantly improve image quality.

## 5 CONCLUSION

In this work, we propose unsupervised discovery of Object Radiance Fields (uORF), which learns to infer object-centric 3D radiance fields from a single image of complex multi-object scenes. We demonstrate uORF's ability on scene segmentation and scene generation in 3D. Our positive results suggest a promising direction to integrate neural rendering into deep probabilistic inference scheme, allowing learning factorized 3D object-centric scene representations from only RGB images.

## ACKNOWLEDGMENTS

This work was in part supported by Qualcomm Innovation Fellowship (QIF), Stanford Institute for Human-Centered AI (HAI), Stanford Center for Integrated Facility Engineering (CIFE), Toyota Research Institute, a Vannevar Bush faculty fellowship, Amazon, Autodesk, Google, and Bosch.

## REPRODUCIBILITY STATEMENT

To ensure reproducibility of our work, we have provided the training and test code repository[†], together with all three synthetic datasets, and pre-trained models on all three datasets. We have also provided a detailed instruction on using our code as well as training on new datasets. In Appendix B, we describe details for re-implementing our work.

## ETHICS STATEMENT

Learning object-centric scene representations is a long-standing topic in vision and it finds various applications in downstream tasks. We represent a 3D scene as a composition of simple radiance fields, which only models object appearances and entangles their physical properties that may be crucial to downstream tasks in a non-interpretable way. However, we envision that careful designs in more structured 3D object representations for specific downstream applications could help improve transparency and human interpretability in model prediction and behavior, allowing both better performances and secure, fair usage. In our code release, we will explicitly specify allowable uses of our system with appropriate licenses. We will use techniques such as watermarking to identify and label visual contents generated by our system.

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

# A  SUPPLEMENTARY MATERIAL OVERVIEW

In the following **supplementary document**, we first provide implementation details on unsupervised discovery of Object Radiance Fields in Section B. We then describe details on datasets and baseline architectures in Section C. We show additional results in in Section D, including results on generalization to unseen object shapes, a demonstration on real photos, an analysis on the sensitivity to slot initialization, and additional qualitative results on all experiments of the main paper and failure cases. In Section E, we show comparison to GIRAFFE (Niemeyer & Geiger, 2020) to demonstrate that it focus on a fundamentally different problem (unconditional generation) than our work (conditional inference). All mathematical and algorithmic notations are the same as those in the main manuscript.

In the **supplementary video**, we provide an overview of our paper.

# B  IMPLEMENTATION

Here, we provide implementation details of our uORF model.

## B.1  OBJECT-CENTRIC LATENT INFERENCE

We show a pseudo code of inferring object-centric latents with the background-aware slot attention in Algorithm 1.

**Convolutional feature extraction.**    The convolutional net extracts features from the input image for updating the latent slots. Our convolutional encoder is a simple U-net. We show our encoder architecture in Table 8 and Table 9. Since we want the model to generalize to decompose unseen images, it is natural to represent foreground objects position and pose in the viewer coordinate system. As identified in previous studies (Tatarchenko et al., 2019), this facilitates the learning of 3D object position and helps generalization. In order for the object-centric representations to include such information in the viewer coordinate system, we can inform the encoder of position information by feeding pixel coordinates and viewer-space ray directions as additional input channels. In our experiments we assume fixed camera focal length. In this case, the ray direction does not provide additional information to the pixel coordinates, and thus we only feed pixel coordinates as input channels in addition to the input RGB image. Each of the $XY$ pixel coordinates is normalized to $[-1, 1]$ in both directions, leading to 4 additional channels to RGB.

## B.2  COORDINATE SPACE AND LOCALITY CONSTRAINT FOR BETTER FORE-/BACK-GROUND DISENTANGLEMENT

**Coordinate space.**    We represent foreground objects in the viewer space. Regarding background environment, we represent it in the world coordinate space for two reasons. Firstly, since it is difficult to estimate full geometry from a single view (e.g., the geometry behind the camera), our model assumes a similar background geometry across scenes and aggregates information about background geometry from multiple sparse views. Representing background in a fixed world space facilitates this aggregation process and empirically leads to better performance. We show a quantitative comparison in Table 10, Table 11 and a visual comparison in Figure 7. Modeling the background in world space provides more details than modeling it in viewer space. Incorporating multi-view images as inference input might relax this assumption (Yu et al., 2020), but we leave it as future exploration.

| Layer name | Input shape | Output shape | Stride | Note |
|---|---|---|---|---|
| Conv1 | 64×64×7 | 64×64×64 | 2 | Skip to Conv6 |
| Conv2 | 64×64×64 | 32×32×64 | 2 | Skip to Conv5 |
| Conv3 | 32×32×64 | 16×16×64 | 2 | |
| Conv4 | 16×16×64 | 16×16×64 | 1 | |
| Upsample | 16×16×64 | 32×32×64 | | Bilinear upsampling |
| Conv5 | 32×32×128 | 32×32×64 | 1 | |
| Upsample | 32×32×64 | 64×64×64 | | Bilinear upsampling |
| Conv6 | 64×64×128 | 64×64×64 | 1 | |

Table 8: Encoder architecture for the CLEVR-567 dataset and the Room-Chair dataset. All convolutional kernel sizes are 3×3. All activation functions for convolutional layers are ReLU.

| Layer name | Input shape | Output shape | Stride | Note |
|---|---|---|---|---|
| Conv0 | 128×128×7 | 128×128×64 | 1 | |
| Conv1 | 128×128×64 | 64×64×64 | 2 | Skip to Conv6 |
| Conv2 | 64×64×64 | 32×32×64 | 2 | Skip to Conv5 |
| Conv3 | 32×32×64 | 16×16×64 | 2 | |
| Conv4 | 16×16×64 | 16×16×64 | 1 | |
| Upsample | 16×16×64 | 32×32×64 | | Bilinear upsampling |
| Conv5 | 32×32×128 | 32×32×64 | 1 | |
| Upsample | 32×32×64 | 64×64×64 | | Bilinear upsampling |
| Conv6 | 64×64×128 | 64×64×64 | 1 | |

Table 9: Encoder architecture for the Room-Diverse dataset. All convolutional kernel sizes are 3×3. All activation functions for convolutional layers are ReLU.

Secondly, this design also encourages the disentanglement between foreground objects and background by preventing the background slot from decoding foreground objects, because the positional information provided in the encoder is represented in viewer space.

**Foreground locality.** To further encourage the disentanglement, we add a locality constraint during early training to prevent foreground slots to represent the background environment. Specifically, considering that "foreground" objects should be largely visible in sight, we set a foreground box and enforce that every foreground-querying point outside the box has zero density. The foreground box is defined such that its projection in image space can engage roughly $90\%$ pixels. The locality constraint is imposed for the first 100K iterations, and it empirically helps prevent the foreground slots from fitting the background. We show a visual comparison in Figure 8, which from we can observe that the model without foreground locality design attaches some background segments to each object.

### B.3 NEURAL RADIANCE FIELD ARCHITECTURE.

We show our conditional object radiance field architecture in Figure 9.

### B.4 MODEL LEARNING

**Loss functions.** We set $\lambda_{\text{percept}} = 0.006$, $\lambda_{\text{adv}} = 0.01$, $\lambda_R = 10$. For perceptual loss, we implement the feature extractor $p$ by using the output of the 4-th convolutional block in a VGG16 (Simonyan & Zisserman, 2014) pretrained on ImageNet. For the adversarial discriminator, we follow the architecture of StyleGAN2 (Karras et al., 2020) with slight modification such that the maximum channel number is 128. We also use the lazy R1 regularization (Karras et al., 2020). We use Adam optimizer for discriminator with learning rate 0.001, $\beta_1 = 0$ and $\beta_2 = 0.9$. The adversarial loss is incorporated after 100K iterations. Since shape uncertainty only appears in the Room-Diverse

---

**Algorithm 1:** Object-centric latent inference with background-aware slot attention.

---

**Input**: feat $\in \mathbb{R}^{N \times D}$
**Learnable**: $\mu^b, \sigma^b, \mu^f, \sigma^f$: prior parameters, $k, q^b, q^f, v^b, v^f$: linear mappings, GRU$^b$, GRU$^f$, MLP$^b$, MLP$^f$

  slot$^b \sim \mathcal{N}^b \in \mathbb{R}^{1 \times D}$           // Sampling slots from priors.
  slots$^f \sim \mathcal{N}^f \in \mathbb{R}^{K \times D}$
  **for** $t = 1, \cdots, T$
   slot_prev$^b$ = slot$^b$,   slots_prev$^f$ = slots$^f$
   attn = Softmax $\left( \frac{1}{\sqrt{D}} k(\texttt{feat}) \cdot \begin{bmatrix} q^b(\texttt{slot}^b) \\ q^f(\texttt{slots}^f) \end{bmatrix}^T, \texttt{dim='slot'} \right)$   // Binding slots to object features.
   attn$^b$ = attn[0],   attn$^f$ = attn[1:end]
   updates$^b$= WeightedMean(weights=attn$^b$, values=$v^b$(inputs)) // Aggregating update signals.
   updates$^f$= WeightedMean(weights=attn$^f$, values=$v^f$(inputs))
   slot$^b$ = GRU$^b$(state=slot_prev$^b$, inputs=updates$^b$)     // Updating slots.
   slots$^f$ = GRU$^f$(state=slots_prev$^f$, inputs=updates$^f$)
   slot$^b$+ = MLP$^b$(slot$^b$),   slots$^f$+ = MLP$^f$(slots$^f$)    // Residual update.
  **return**   slot$^b$, slots$^f$

---

| Models | LPIPS↓ | SSIM↑ | PSNR↑ |
|---|---|---|---|
| uORF w/ view-space Backg. | 0.151 | 0.799 | 27.86 |
| uORF (ours) | **0.082** | **0.872** | **29.60** |

Table 10: Ablation for background coordinate space on novel view synthesis on Room-Chair dataset.

| Models | 3D metric | 2D metric | |
|---|---|---|---|
| | NV-ARI↑ | ARI↑ | Fg-ARI↑ |
| uORF w/ view-space Backg. | 73.5 | 78.0 | **89.0** |
| uORF (ours) | **74.3** | **78.8** | 88.8 |

Table 11: Ablation for background coordinate space on segmentation on Room-Chair dataset.

dataset, we only impose the adversarial loss on the Room-Diverse dataset but not on CLEVR-567 or Room-Chair. Both perceptual loss and adversarial loss are added after the first 100K iterations.

**Coarse-to-fine progressive training.** For coarse training, we bilinearly downsample supervision images to 64×64. The coarse training lasts for 600K iterations. For fine training, we randomly crop 64×64 patches from 128×128 images. The fine training lasts for 600K iterations. Our model is trained on a single Nvidia RTX 3090 GPU for about 6 days. For all networks except discriminator, we use Adam optimizer with learning rate 0.0003, $\beta_1 = 0.9$ and $\beta_2 = 0.999$. Learning rate is exponentially decreased by half for every 200K iterations until after 600K iterations. We also adopt the learning rate warm-up from the slot attention paper (Locatello et al., 2020) for the first 1K iterations. We initialize decoder networks with Xavier's initialization. In each batch, we input one image and neurally render 4 images for supervision. We render each pixel with 64 samples.

## C   EXPERIMENTS

In this section we provide further details on experiment settings.

### C.1   DATA

For the construction of all three datasets, the training/testing sets share the same pool of textures, shapes, and colors. The scenes in both sets differ in the spatial arrangement of objects, as well as the appearance differences induced by soft shadows and inter-reflections due to global illumination effects.

**CLEVR-567.** In the CLEVR-567 dataset, each object's shape is randomly chosen from three geometric primitives (i.e., cylinder, cube and sphere). The color is randomly chosen from {red, blue, purple, gray, cyan, yellow, green, brown}. There are two possible sizes for each object. When rendering images, we use the same camera intrinsic as original CLEVR dataset (Johnson et al., 2017). We do not use the visibility check due to our 360 degree multi-view setting, so we increase elevation angle by $\pi/15$ to increase the chance of object visibility. Rendering setting is the same for all datasets.

For CLEVR-567 dataset we set the latent dimension $D = 40$ and the maximum number of objects $K = 8$.

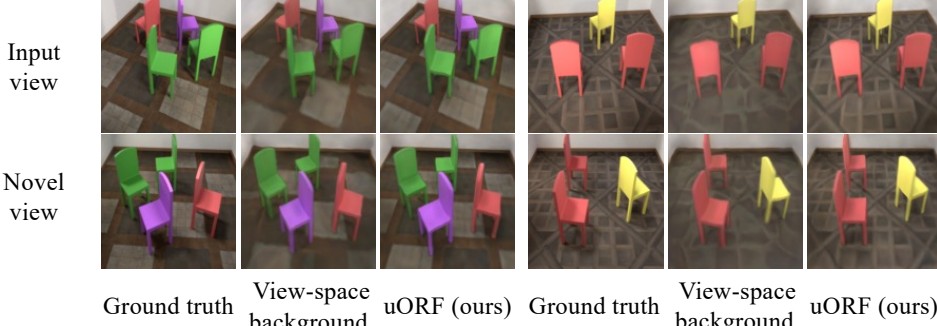

| Ground truth | View-space background | uORF (ours) | Ground truth | View-space background | uORF (ours) |

Figure 7: Visual comparison for representing background on view-space on novel view synthesis.

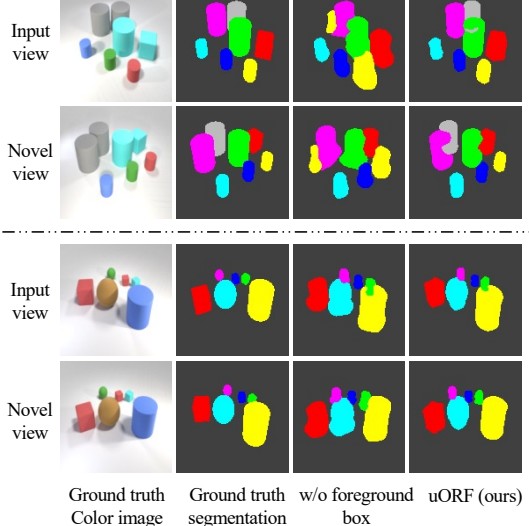

| Ground truth Color image | Ground truth segmentation | w/o foreground box | uORF (ours) |

Figure 8: Visual comparison on ablation for foreground locality constraint. We show examples in CLEVR-567 testset. We can see that our foreground locality box helps prevent object slots from fitting background segments.

**Room-Chair.** For the object shape we use a chair model[‡] from ShapeNet (Chang et al., 2015). We use the same material and colors as CLEVR-567. For Room-Chair and Room-Diverse datasets, we set the latent dimension $D = 64$ and the maximum number of objects $K = 5$.

**Room-Diverse.** All object shapes are randomly chosen from 1,200 ShapeNet chairs. For each shape, we normalize it into a unit cube according to vertex coordinates. We also use 8 colors {red, blue, purple, gray, cyan, yellow, green, black} with diffuse material. Since shape uncertainty only appears in this dataset, we only impose the adversarial loss on this dataset.

### C.2 BASELINE ARCHITECTURES

**Slot attention.** We use the encoder-decoder architecture in the slot attention paper (Locatello et al., 2020) used for object discovery experiments on the CLEVR dataset. Basically it has 6 convolutional layers for encoder and 6 convolution-transpose layers for decoder. The number of channels for each layer is 64. All models are trained on 128×128 images.

**NeRF-AE.** We follow NeRF implementation without view direction as input and set the highest frequency to 5. The encoder is similar to ours in Figure 9, but the basic number of channels is increased from 64 to 256 (and thus the number of channels of inputs to Conv5 and Conv6 is 512). The number of slot is set to 1.

## D ADDITIONAL RESULTS

**Generalization to unseen objects.** In the main paper we demonstrate systematic generalization to unseen combination of shape and color, here we further validate our model's generalization to unseen

---

[‡]Model ID: 3ffd794e5100258483bc207d8a5912e3

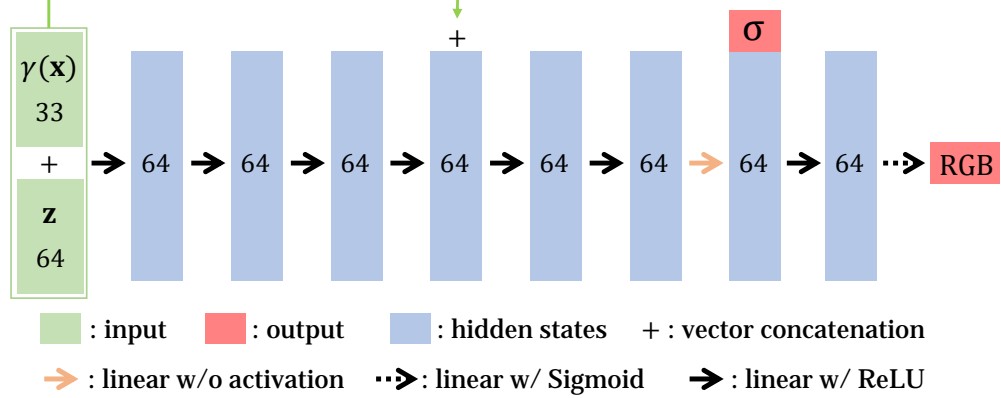

Figure 9: Illustration for foreground decoder architecture. We follow the architecture in NeRF (Mildenhall et al., 2020) but with fewer parameters to decrease space demand. We set the highest positional embedding frequency to 5, so that the positional embedding input dimension is $5 \times 2 \times 3 + 3 = 33$. The background decoder is slightly different in that it does not have the second last layer and third last layer. Density $\sigma$ is activated by ReLU. Since estimating specularity from a single image is intrinsically ambiguous, we assume Lambertian surfaces and do not use the ray direction as input.

| Models | LPIPS↓ | SSIM↑ | PSNR↑ |
|---|---|---|---|
| uORF on seen shape testset | **0.1729** | 0.7094 | 25.96 |
| uORF on unseen shape testset | 0.1771 | **0.7125** | **26.16** |

Table 12: Novel view synthesis results on unseen/seen shape testset of Room-Diverse.

| Models | 3D metric | 2D metric | |
|---|---|---|---|
| | NV-ARI↑ | ARI↑ | Fg-ARI↑ |
| uORF on seen shape testset | 56.9 | 65.6 | **67.9** |
| uORF on unseen shape testset | **57.0** | **66.1** | 67.7 |

Table 13: Unsupervised segmentation in 3D results on unseen/seen shape testset of Room-Diverse.

object shapes. To this end, we construct another test set for Room-Diverse. All test objects in the new test set are drawn from a pool of 500 shapenet chairs that are completely disjoint from the 1200 training chairs. All other settings are the same as the original test set. We show quantitative results in Table 12 for novel view synthesis and in Table 13 for segmentation. As we can see, our model yields the same level of performances even on the unseen shape test set, suggesting its generalization to unseen object shapes.

**Generalization to real images.** We also take a step further to test our pretrained model's generalization on real photos. To do this, we use uORF trained on Room-Diverse. We take a few real photos by a cellphone, providing an input image and a few reference images. We show the visual results in Figure 10. Although the real photo has a different imaging process and consists of unseen objects and background, uORF is able to discover all objects with roughly correct positions and orientations, yielding plausible segmentation results and object-moving results.

**Analysis on the sensitivity to slot initialization.** We test the robustness of our model to the slot initialization on the Room-Chair dataset. For each test scene, we now use 5 different random seeds for sampling initial centers. We compute the mean $\mu$ and std $\sigma$ of ARI over the 5 seeds. We average them over the 500 test scenes. The averaged mean $\bar{\mu}$ of ARI is 78.8% and $\bar{\sigma}$ is 1.7%. The mean ARI suggests good segmentation results (very close to 78.8% as reported in Table 2 in our main paper), and $\bar{\sigma} = 1.7\%$ indicates that different seeds all lead to results close to such good ARI performance.

**Additional qualitative results.** We show additional qualitative results for our experiments in the main manuscript. We show additional examples for scene segmentation in Figure 11 and Figure 12, for novel view synthesis in Figure 13, Figure 14 and Figure 15, for scene editing in Figure 16 and Figure 17, for evaluating losses in Figure 18, for generalization to challenging spatial arrangement in Figure 19 (note that in the packed-CLEVR-11 dataset we only use a single size for higher object visibility), and for generalization to unseen object appearance in Figure 20.

**Failure case.** In our experiments, we observed a type of failure which we call "attention rank-collapse". We show examples in Figure 21.

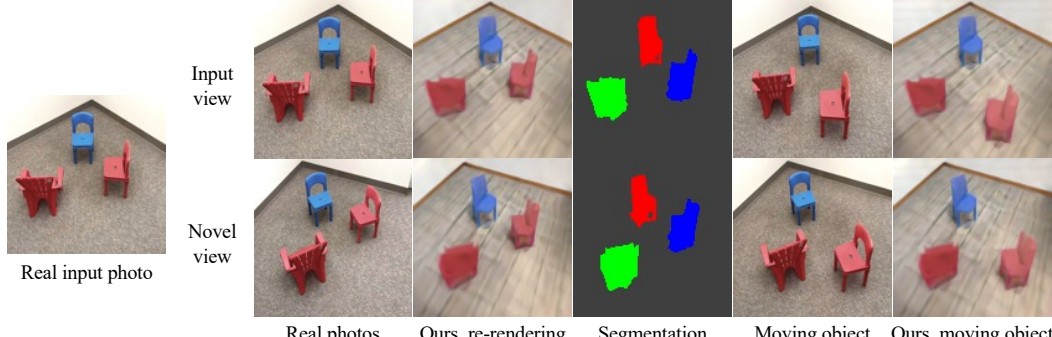

Figure 10: Demonstration on generalization to real photos. We use uORF pretrained on Room-Diverse and take photos by a cellphone.

"Attention rank-collapse" refers to that all the foreground object slots have (nearly) the same attention map and collapse to the same representation. Each collapsed slot decodes simply nothing (or all the foreground objects). This "attention rank-collapse" happens when the initialization is prompt to a degenerate solution for the slot attention. It occasionally happens and empirically changing the initialization seed can address it. A related rank-collapse problem is discussed in Dong et al. (2021), which suggests that adding some architectural inductive bias can largely alleviate the problem. We hope future research can address this problem fundamentally.

## E    COMPARISON TO GIRAFFE

Our work has a fundamentally different focus compared to GIRAFFE (Niemeyer & Geiger, 2020). While GIRAFFE focuses on unconditional generation and enables multi-object scene synthesis and rendering, the goal of our uORF is to simultaneously infer 3D multi-object scene representations from a single image, in addition to using those representations for rendering and editing as in GIRAFFE.

### E.1    COMPARISONS BETWEEN OUR uORF AND GIRAFFE

To demonstrate that the inference of such multi-object scenes is highly non-trivial, we compare with GIRAFFE on both CLEVR-567 and Room-Chair (we cannot compare on their datasets because they only have a single image for each scene). To train GIRAFFE on our datasets, we use the official repo§ and the same hyper-parameters that GIRAFFE authors used for their CLEVR-2345 dataset, except for a few adaptive changes to our datasets: (1) We try different sizes for the object slot, because CLEVR-2345 only uses small objects while our datasets both contain larger objects. Specifically, we try $2\times$, $1.5\times$, and $1\times$ original size, and use the one with the lowest FID for each dataset. (2) We adjust the camera elevation angle and focal lengths to match our datasets. (3) We set the number of objects to 4 for the Room-Chair dataset because each scene has no more than 4 chairs. We train the GIRAFFE models for around 500K iterations on 128-by-128 images, such that FID does not drop anymore.

For GIRAFFE to do inference, we sample object (including background) latents and positions in the same manner as training, and then we optimize for L2 reconstruction loss for both the latents and the positions. We use Adam and do a learning rate sweep to select the one that leads to the best reconstruction loss. We divide the learning rate by 10 when the loss plateaus. We do this learning rate decay twice. We sweep in $\{0.1, 0.01, 0.001, 0.0001\}$ and find that $0.01$ works best. Since each scene has an unknown number of objects, we set the number to the maximum number across all scenes. It converges at around 150 iterations on CLEVR-567 and around 300 iterations on Room-Chair. Thus we set the maximum iteration to 300 and 500 for them, respectively.

We also compare with a GIRAFFE model that is pretrained on CLEVR-2345. The pretrained model is provided by the authors. The pretrained model yields FID= $82$ on CLEVR-567 (FID= $61$ on CLEVR-2345), indicating that it could be a valid baseline even though the two datasets are mildly different.

We show input-view reconstruction and novel view synthesis results in Table 14 and Table 15, and we show qualitative comparison in Figure 22 and Figure 23. We can see that GIRAFFE fails in

---

§https://github.com/autonomousvision/giraffe

| Models | Input view reconstruction | | | Novel view synthesis | | |
|---|---|---|---|---|---|---|
| | LPIPS↓ | SSIM↑ | PSNR↑ | LPIPS↓ | SSIM↑ | PSNR↑ |
| GIRAFFE (trained on CLEVR-567) | 0.330 | 0.815 | 23.75 | 0.549 | 0.672 | 16.65 |
| GIRAFFE (author-pretrained model on CLEVR-2345) | 0.382 | 0.780 | 21.76 | 0.643 | 0.348 | 11.70 |
| uORF (ours) | **0.085** | **0.901** | **29.33** | **0.086** | **0.897** | **29.28** |

Table 14: Inference comparison with GIRAFFE on CLEVR-567.

| Models | Input view reconstruction | | | Novel view synthesis | | |
|---|---|---|---|---|---|---|
| | LPIPS↓ | SSIM↑ | PSNR↑ | LPIPS↓ | SSIM↑ | PSNR↑ |
| GIRAFFE (trained on Room-Chair) | 0.414 | 0.597 | 20.90 | 0.588 | 0.538 | 18.53 |
| uORF (ours) | **0.085** | **0.876** | **29.65** | **0.082** | **0.872** | **29.60** |

Table 15: Inference comparison with GIRAFFE on Room-Chair.

reconstructing the multi-object scenes from a single image, as well as novel view synthesis. Let alone segmentation in 3D.

### E.2  GIRAFFE INFERENCE ON CLEVR-2345

While we have compared with GIRAFFE on our datasets, we further evaluate the author-provided pretrained model on the simpler dataset CLEVR-2345 from the GIRAFFE paper itself. We found that while GIRAFFE does well on unconditional scene synthesis, it cannot perform novel view synthesis on their own dataset, either. This shows that GIRAFFE focuses on problems very different from ours.

We first show that inference/reconstruction is challenging for GIRAFFE, even on the simpler dataset. We do inference on the **author-provided CLEVR-2345 dataset** using the **author-provided pretrained model**. We show randomly sampled examples through the iterative inference process in Figure 24.

Then we show that GIRAFFE fails in wide-baseline novel view synthesis. We use the author-provided pretrained model to sample from its latent space and unconditionally generate one image. Then we keep all the variables the same, but circularly move cameras to render novel views. We show 10 random examples of this circular novel view synthesis in Figure 25. We see that when the viewpoint changes become significant, GIRAFFE fails novel view synthesis, because its neural renderer is based on 2D feature maps and it's not inherently 3D.

### E.3  DISCUSSION AND SUMMARY

In general, inverting GAN latent space even for the holistic image is non-trivial and needs architectural-specific designs (we refer the reader to the discussion and references in a recent survey on GAN inversion (Xia et al., 2021)). As for inverting compositional multi-object scenes, it becomes even harder due to ambiguous correspondences ("which slot corresponds to which object?"), number of objects ("how many slots should I put?"), object position constraints ("there are two objects overlapping in the image, but they should not be overlapping in 3D"), optimization issues (e.g., optimizing rotation is notoriously difficult (Zhou et al., 2019)), etc.

In summary, it is highly non-trivial for GIRAFFE to do inference for multi-object scenes due to complexities such as ambiguous correspondences, the number of objects, and optimization issues. We will include more discussions on the difference between the two methods in the following separate thread. In short, our uORF tries to solve a fundamentally different problem from GIRAFFE, i.e., we aim at inferring the joint distribution of objects from a single image while GIRAFFE targets extrinsic-controllable image generation. Therefore, our method enables novel tasks such as unsupervised segmentation and editing in 3D, where prior methods including GIRAFFE are not able to do.

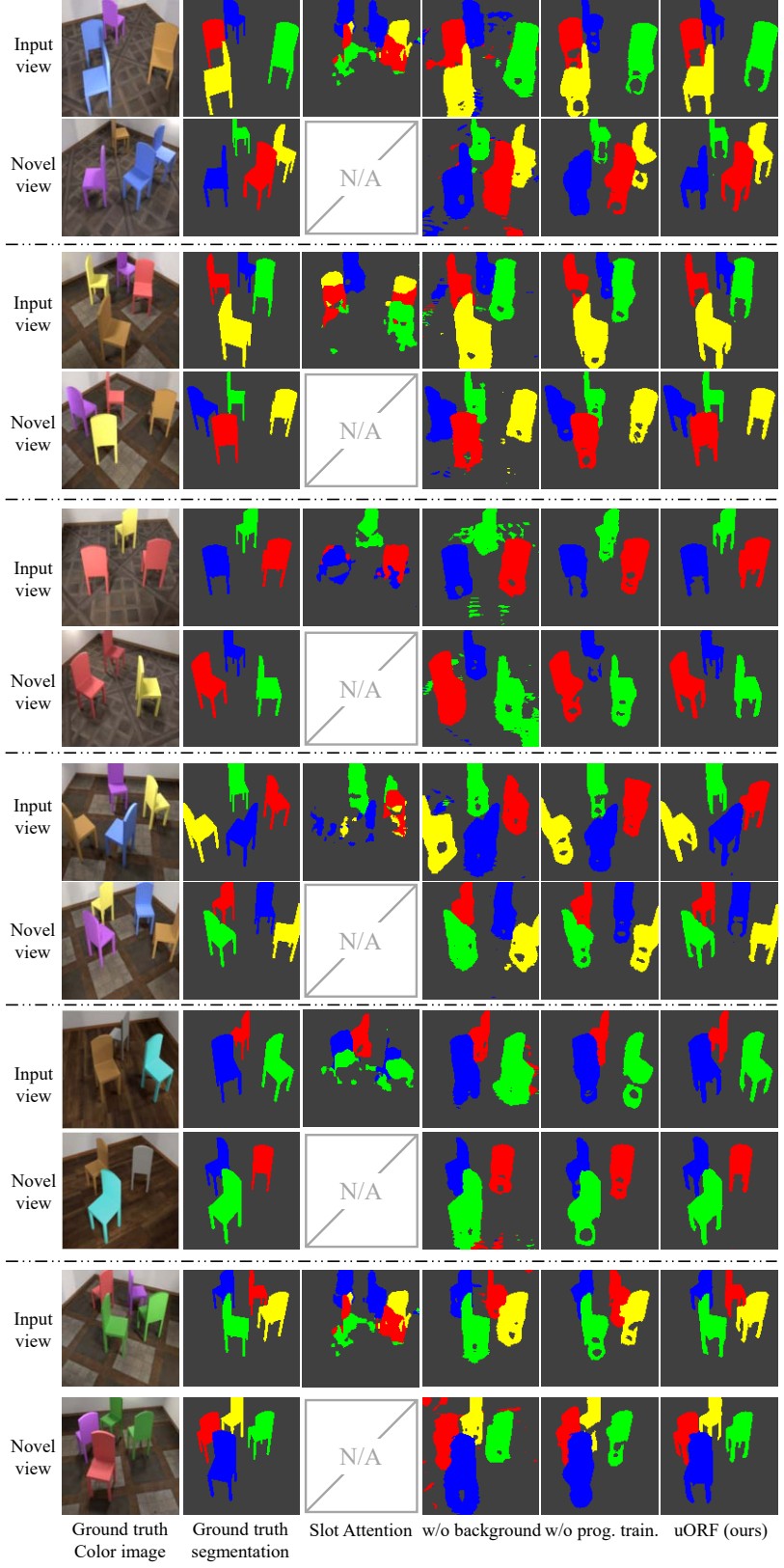

Figure 11: Additional qualitative results for segmentation in 3D on Room-Chair dataset.

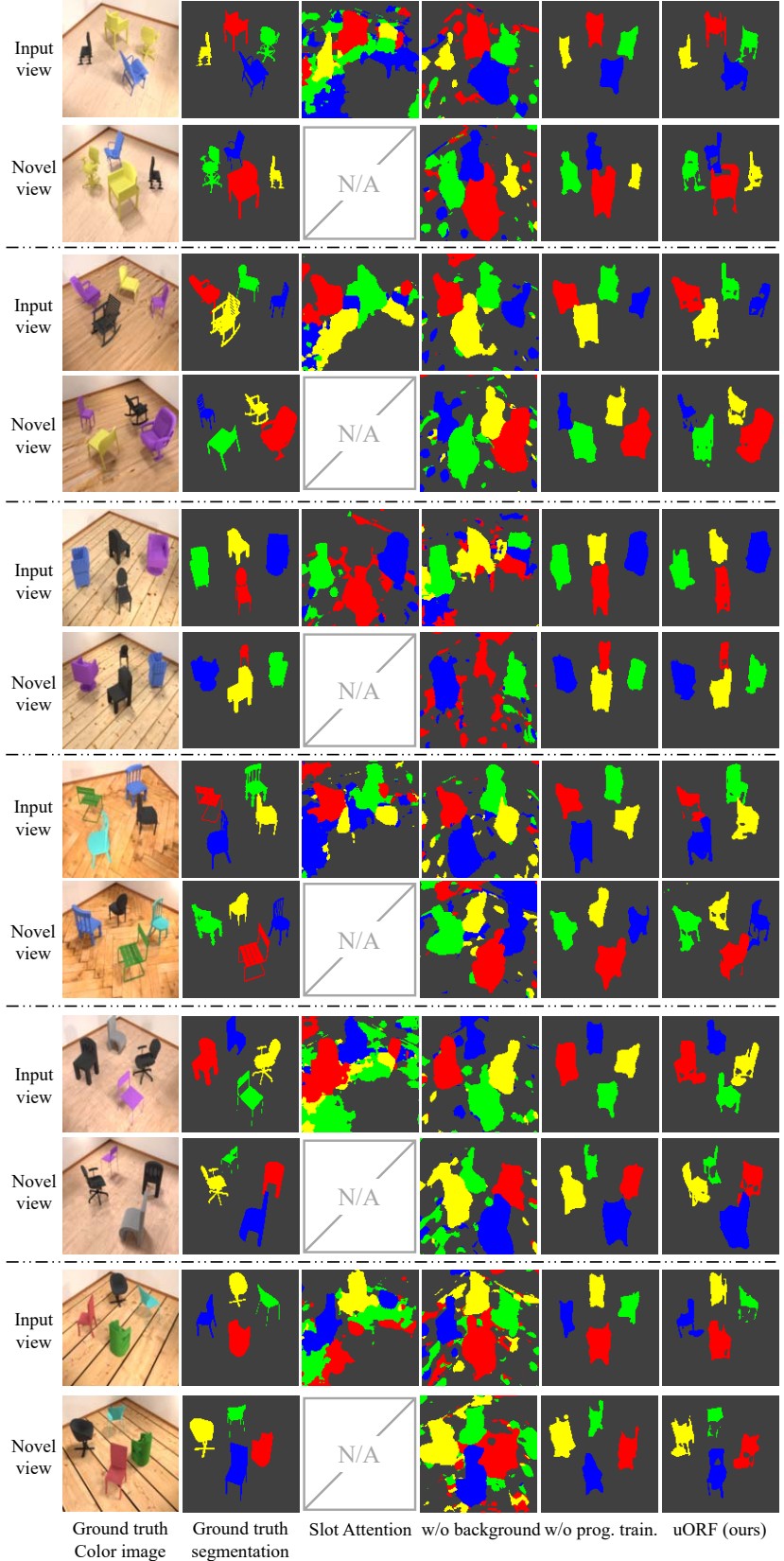

Ground truth   Ground truth   Slot Attention   w/o background   w/o prog. train.   uORF (ours)
Color image    segmentation

Figure 12: Additional qualitative results for segmentation in 3D on Room-Diverse dataset.

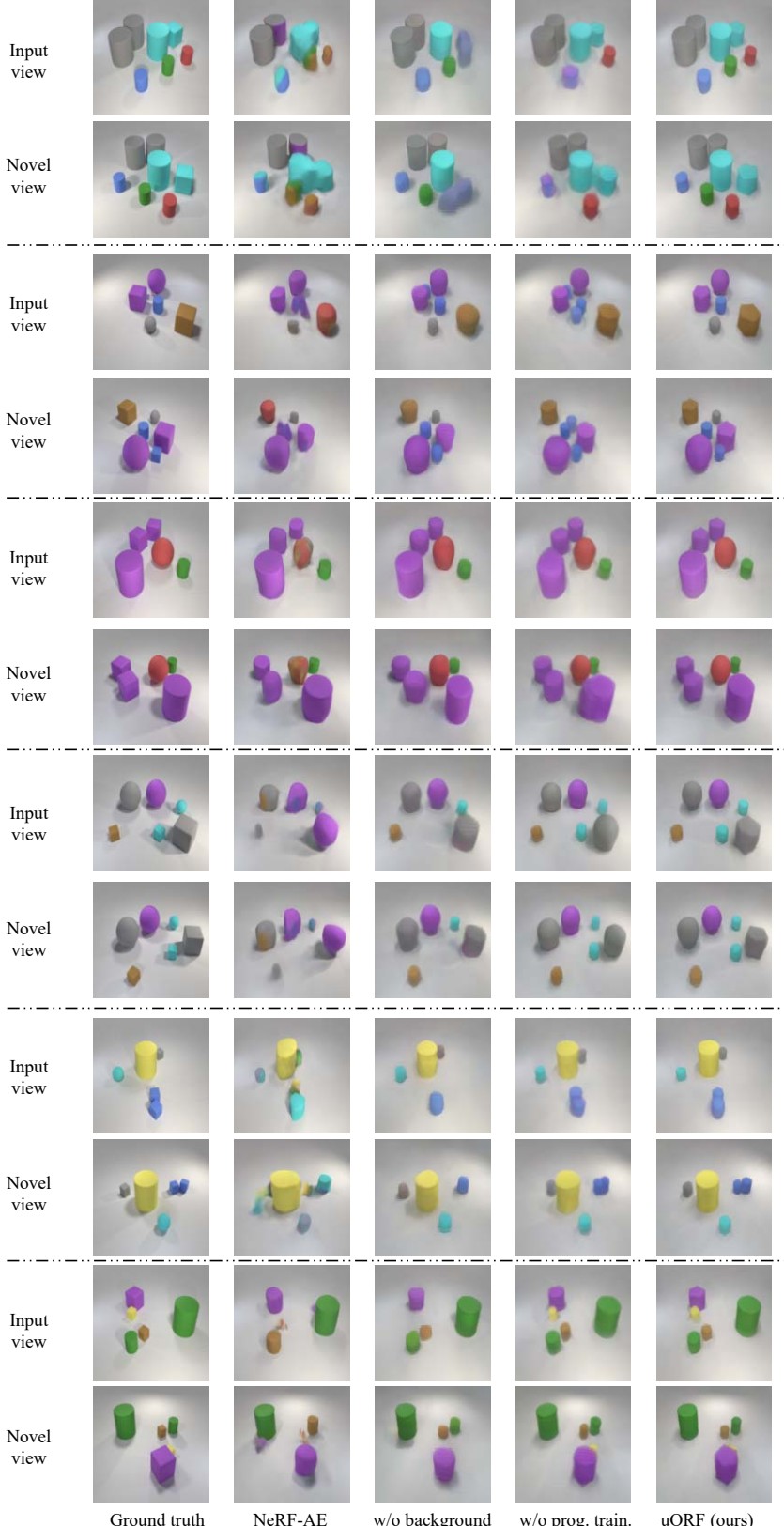

Figure 13: Additional qualitative results for novel view synthesis on CLEVR-567 dataset.

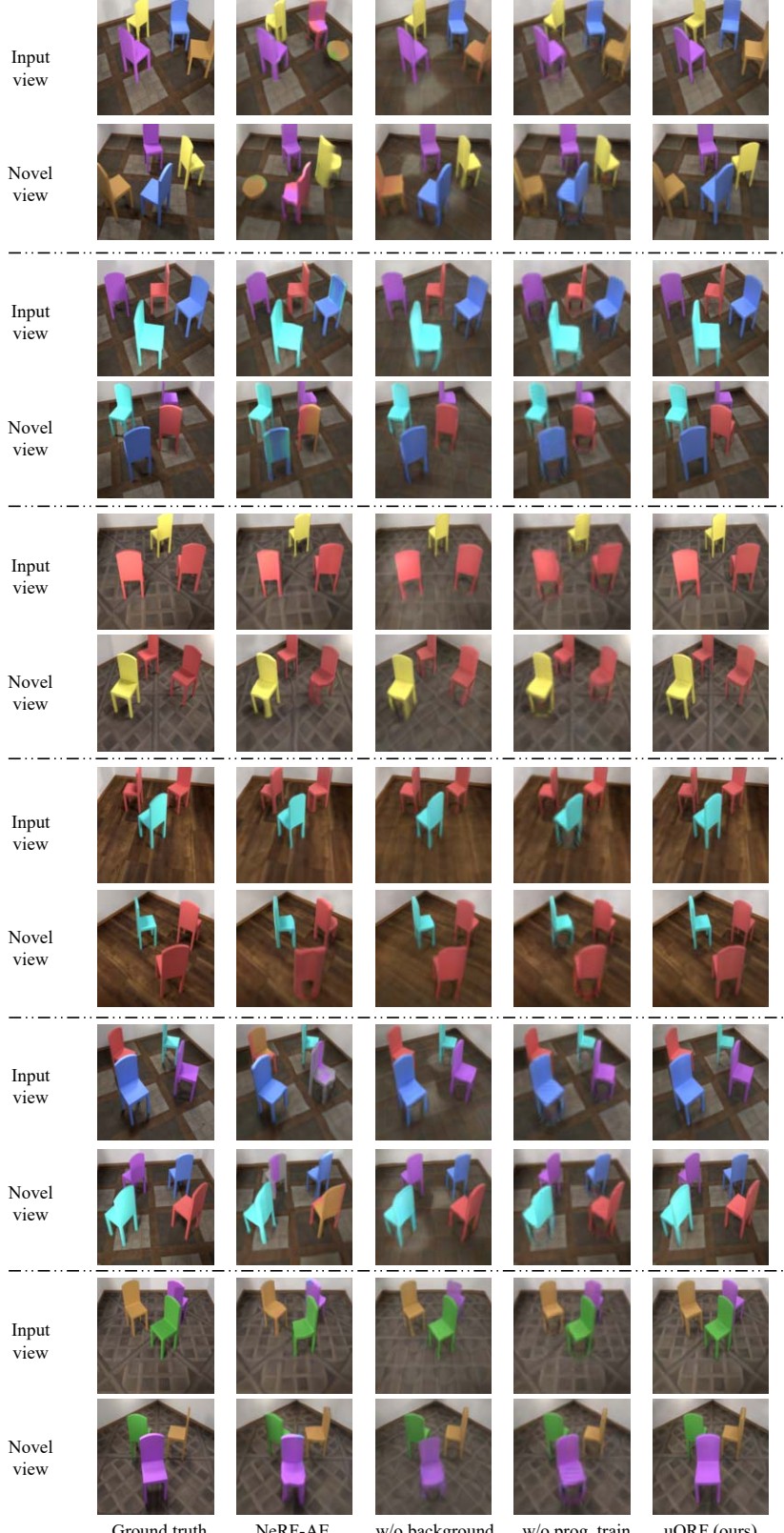

Figure 14: Additional qualitative results for novel view synthesis on Room-Chair dataset.

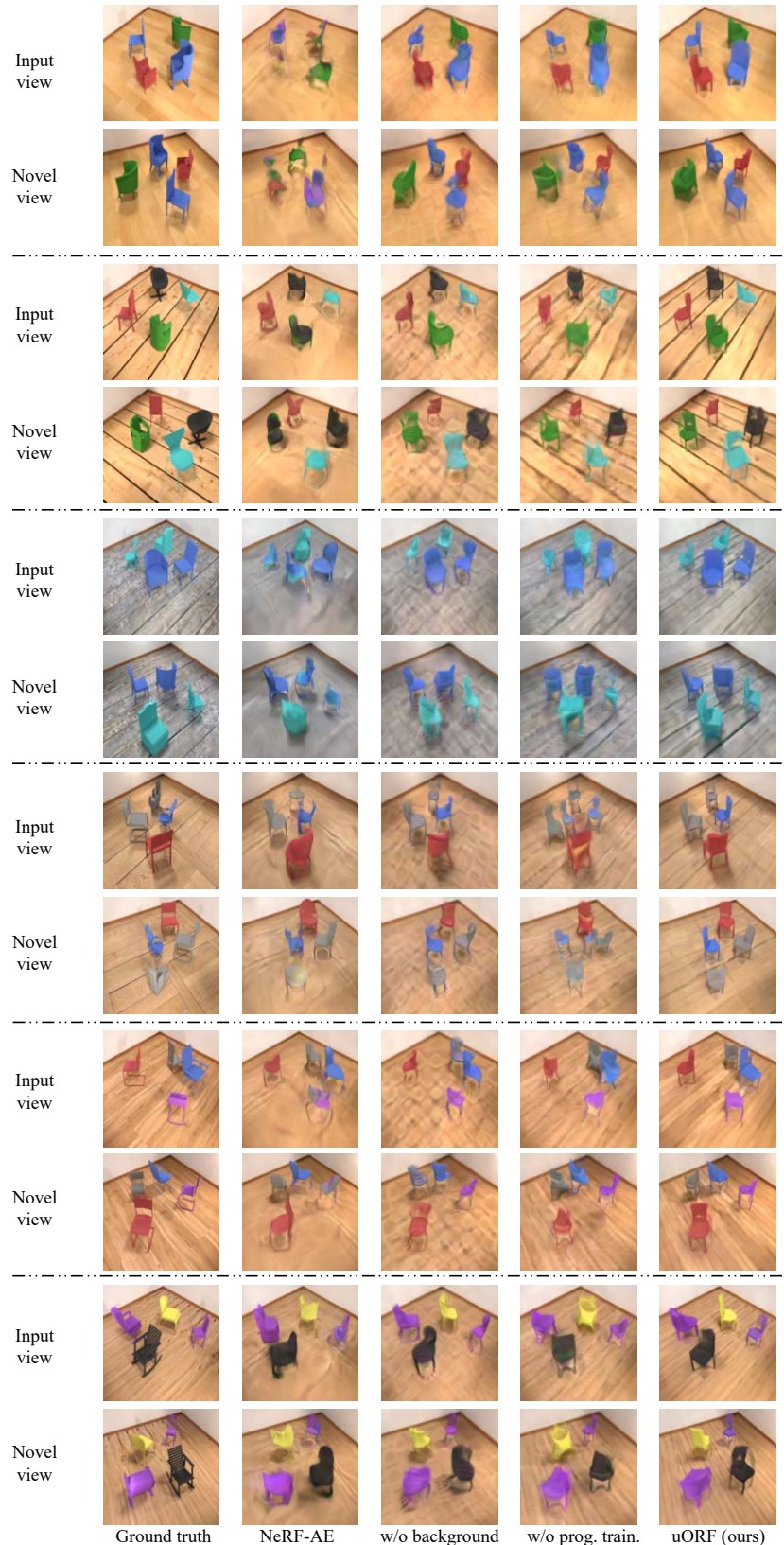

Figure 15: Additional qualitative results for novel view synthesis on Room-Diverse dataset.

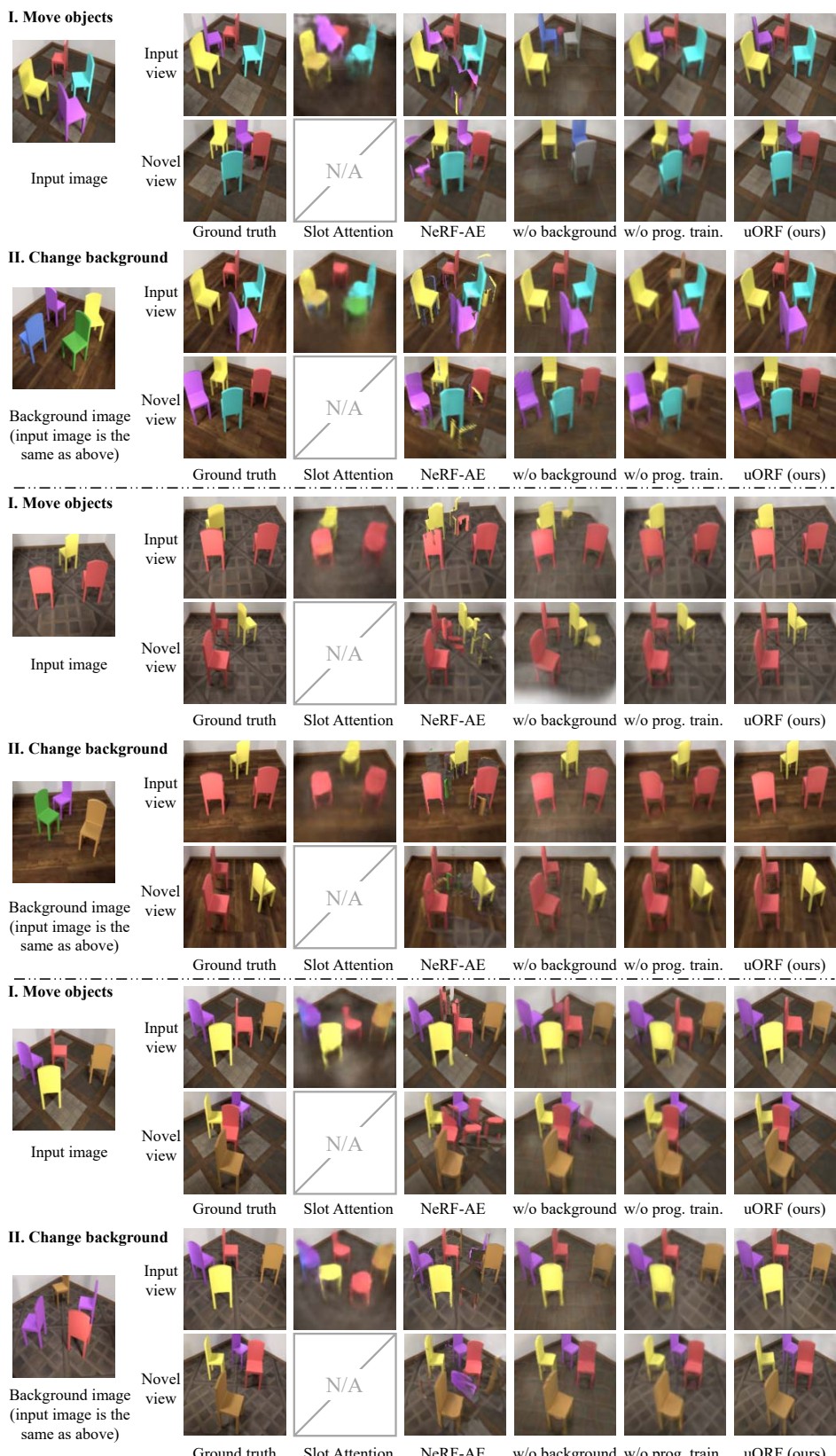

Figure 16: Additional qualitative results for scene editing.

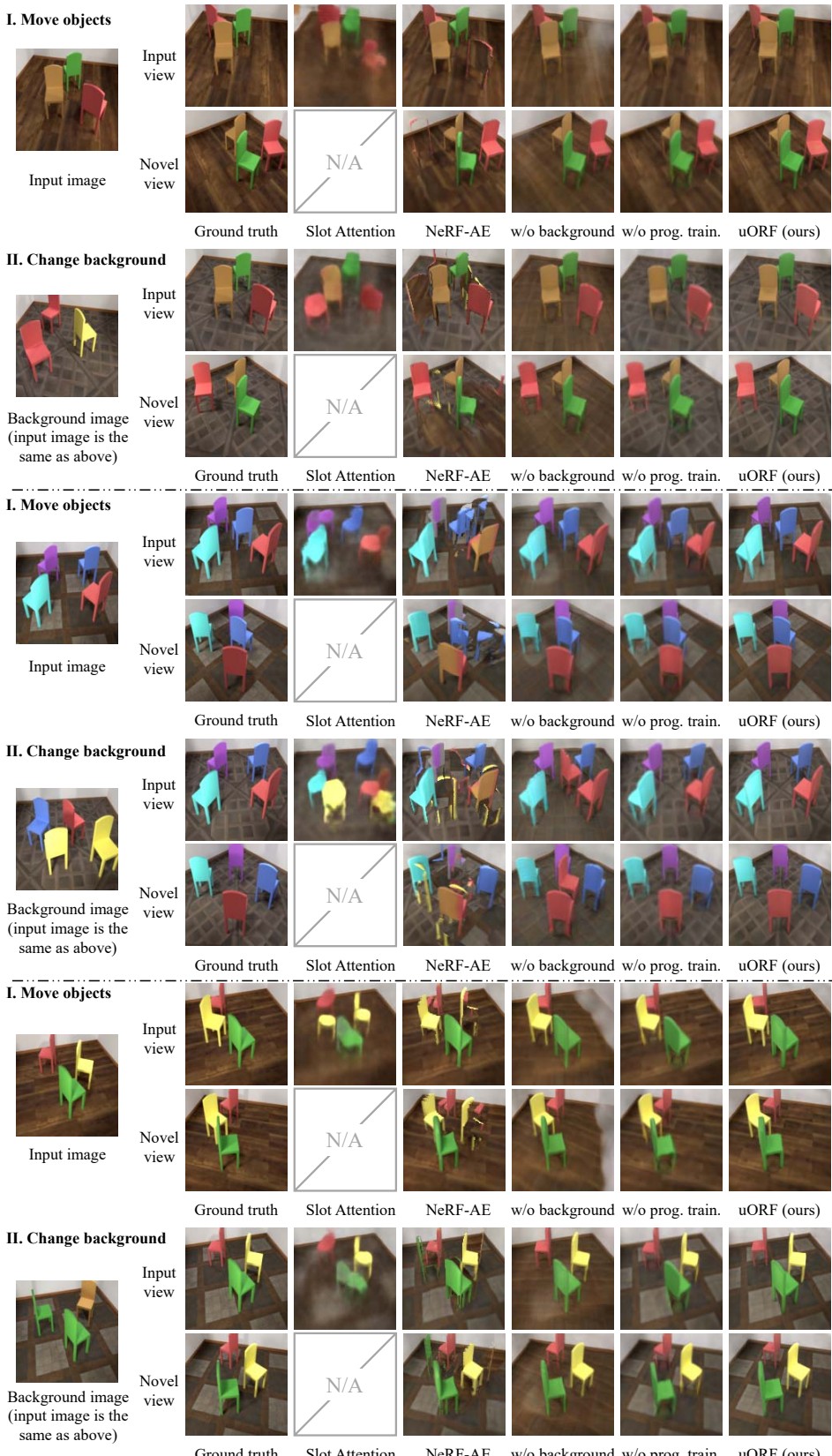

Figure 17: Additional qualitative results for scene editing.

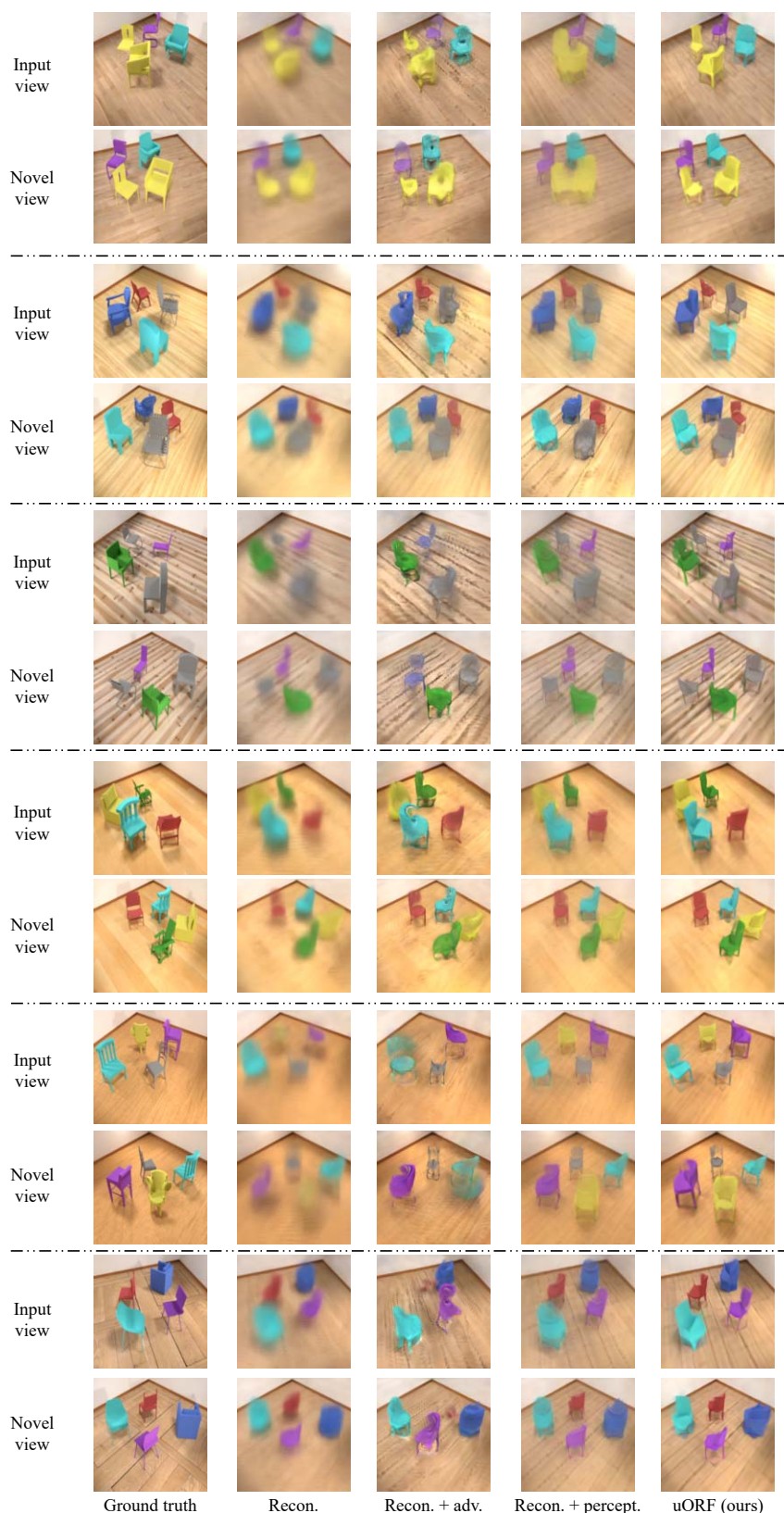

Figure 18: Qualitative results for loss evaluations. Using both perceptual loss and adversarial loss improves image quality.

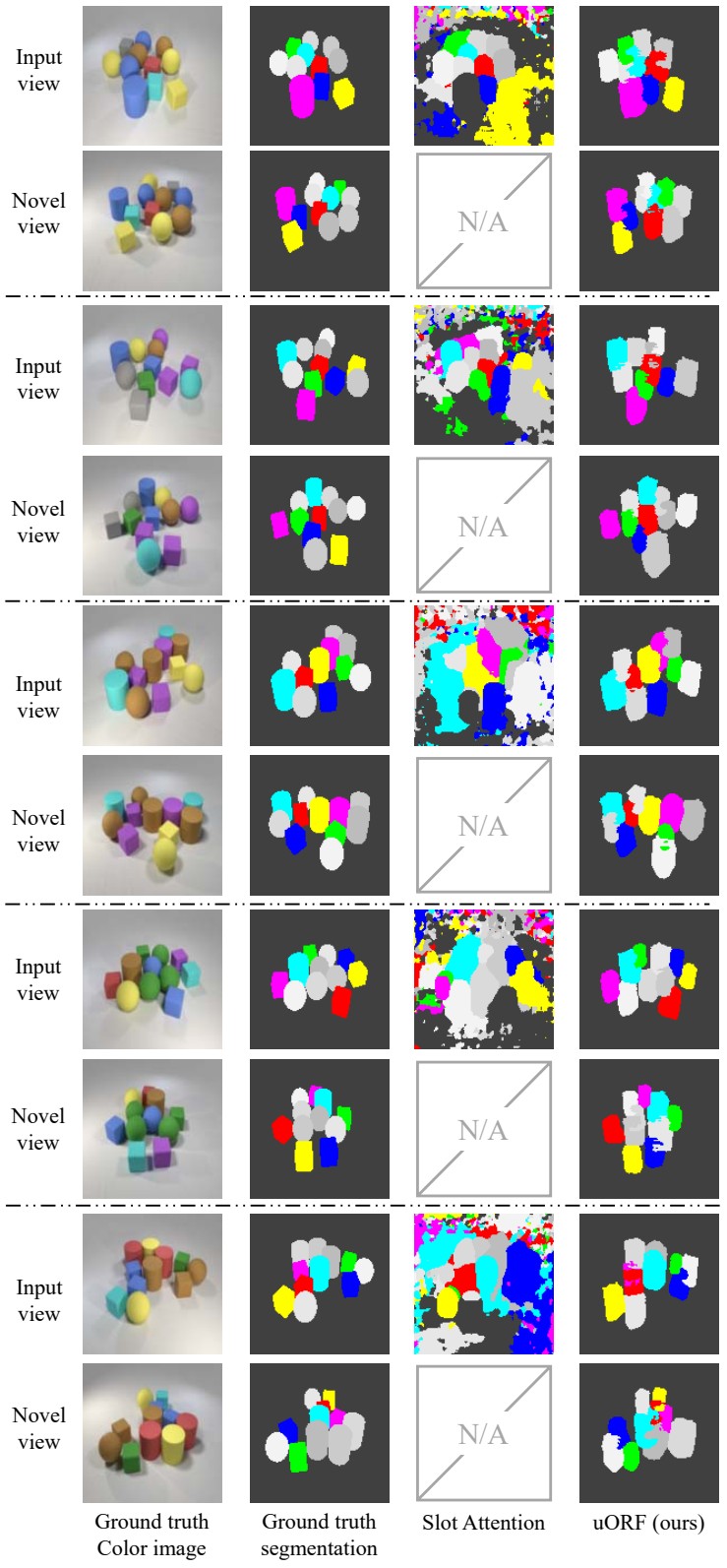

Figure 19: Qualitative results for generalization to unseen spatial arrangement.

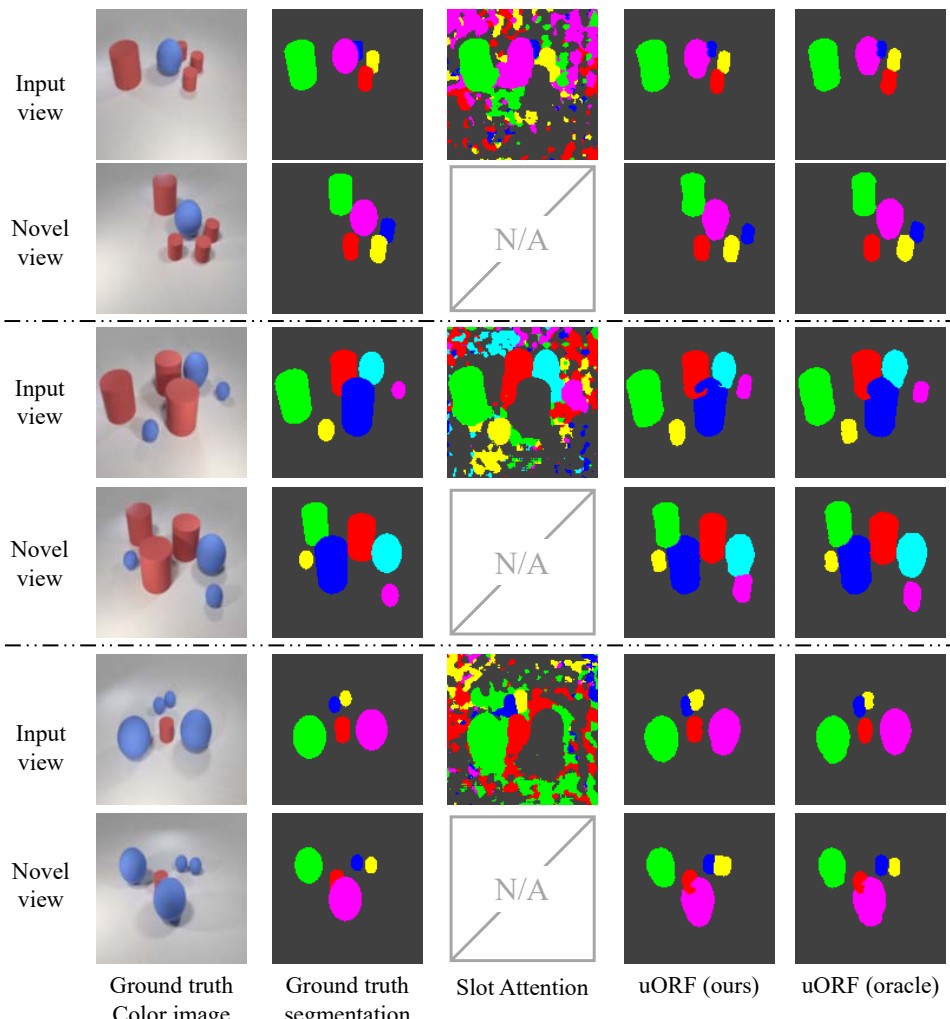

|  | Ground truth Color image | Ground truth segmentation | Slot Attention | uORF (ours) | uORF (oracle) |

Figure 20: Qualitative results for generalization to unseen combination of color and shape.

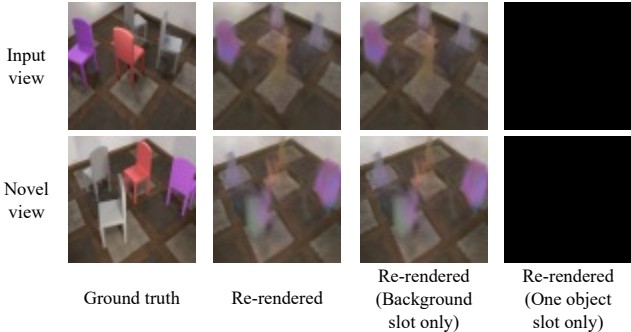

Figure 21: Failure case of our model, which we call "attention rank-collapse". All foreground slots share the same attention map. Every foreground slot decodes to the same radiance field (empty radiance here) rather than specializing to an object. Here we only show one object slot, as all others look the same.

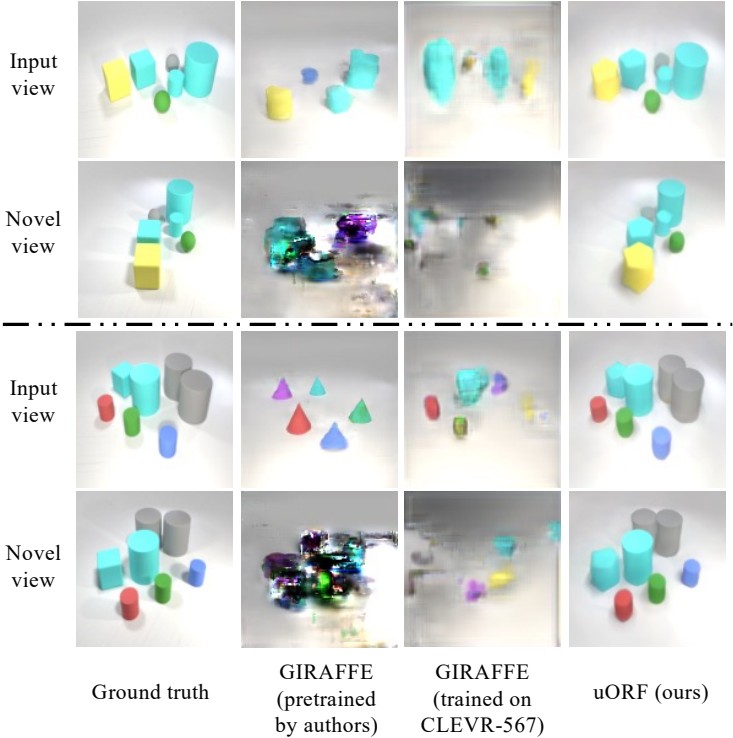

Figure 22: Visual comparison with GIRAFFE for inference on CLEVR-567 dataset. GIRAFFE fails inference.

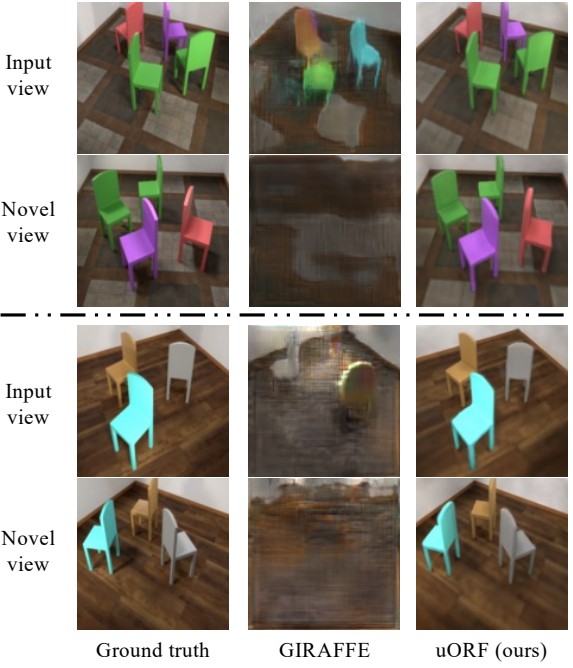

Figure 23: Visual comparison with GIRAFFE for inference on Room-Chair dataset. GIRAFFE fails inference.

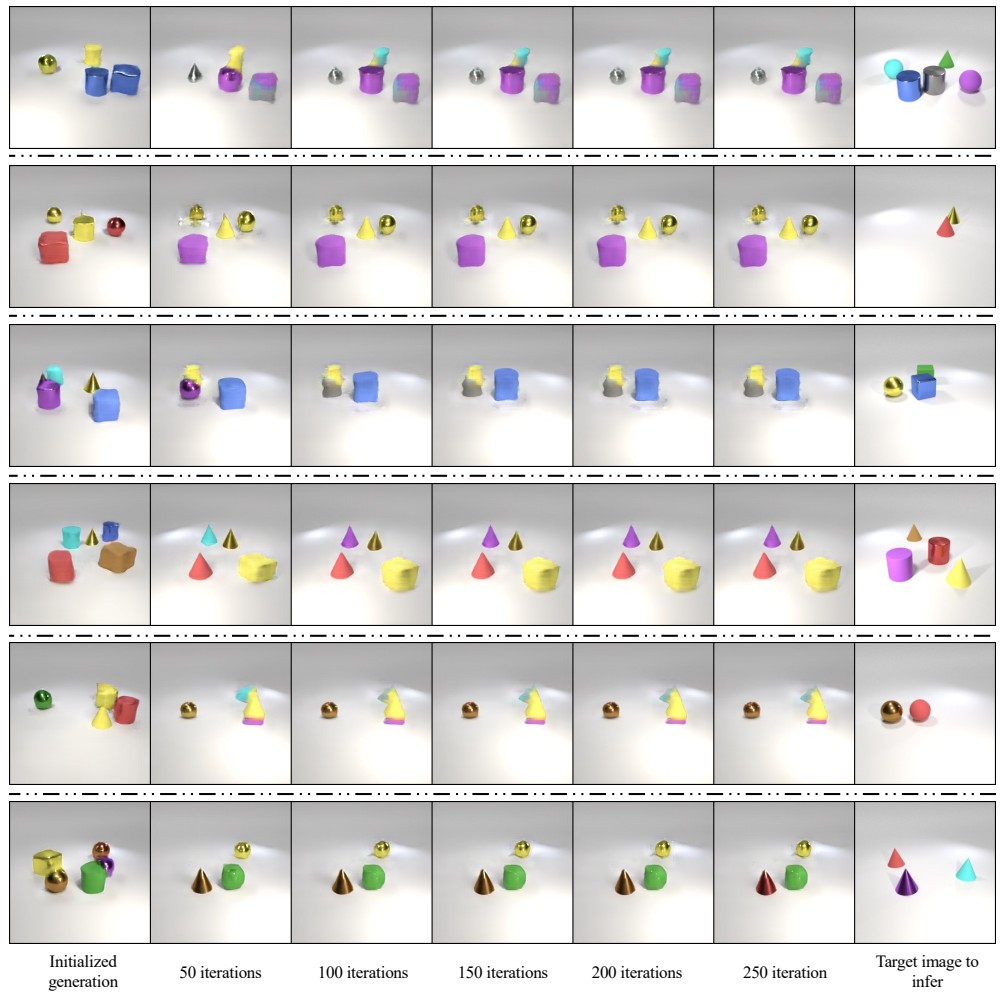

Figure 24: Inference trajectory of GIRAFFE using author-provided models on the author-provided dataset CLEVR-2345.

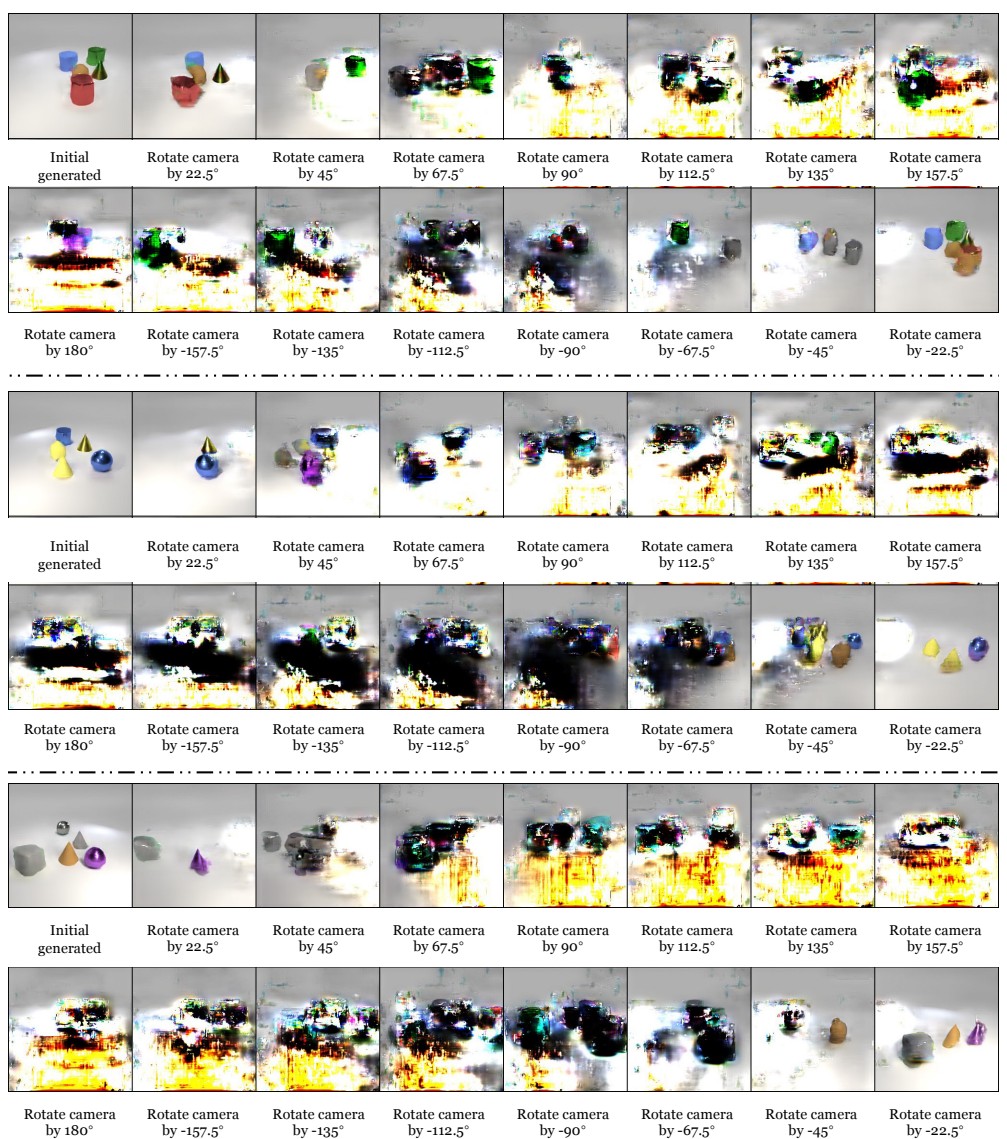

Figure 25: Novel view synthesis on randomly generated examples using author-provided pretrained GIRAFFE model on CLEVR-2345. GIRAFFE fails inference of these multi-object scenes. GIRAFFE cannot synthesize novel views with large rotations.

