# OpenReview forum: "Unsupervised Discovery of Object Radiance Fields"
_ICLR.cc/2022/Conference — ICLR 2022 Poster_

### Official Review · Reviewer_pN74 · 2021-11-01

**Correctness:** 3
**Technical Novelty And Significance:** 3
**Empirical Novelty And Significance:** 4
**Recommendation:** 8
**Confidence:** 4

**Main Review:**

##################################################################
# Pros
1. The authors address a significant new problem: modeling 3D scenes as a disjoint set of objects that can be combined and rendered for novel view synthesis. Their model can also be learned from only 2D data.

2. The proposed method uses state-of-the-art techniques to achieve its goal. Namely, it uses slot-attention mechanisms (NeurIPS 2020) and Neural Radiance Fields (ECCV 2020) and combines them to address a new problem.

3. Treating background and foreground latent vectors as being drawn from two separately learnable distributions addresses one of the significant drawbacks of slot attention.

4. I appreciated the authors mentioning that concurrent work by Stelzner et al. (2021) addresses the same issue and differentiates this paper appropriately.

5. The proposed method is overall technically sound, and code is provided, which will help with reproducibility. Also, the authors do an excellent job at mentioning all the hyper-parameters and model architecture details as far as I can see.

6. The paper provides comprehensive experimental evaluations, both quantitative and qualitative.

####################################################################
# Negatives / Questions
1. It would have been great to compare this work to the concurrent work of Stelzner et al. (2021). I believe that this would help the community to put the two concurrent submissions into context. That said, I do recognize that the work of Stelzner et al. (2021) has not been published and that code/data for their approach is not publicly available, making comparisons extremely difficult and should therefore not be a requirement for publication of this work. I hope that future work in this direction will pick up this issue.

2. Section 3.3, Coarse to fine training. I agree that rendering images with the volumetric rendering framework proposed in NeRF requires many evaluations per ray, so reducing the number of rays sampled during training makes sense. One detail that is missing in this section is how many samples per ray are used? Moreover, do the authors still use the two networks for ray evaluation as NeRF (coarse and fine ones)? If not, why not? Second: In your approach, you sample random patches during the fine training stage and downsample the images during the coarse training stage. In this paper: "GRAF: Generative Radiance Fields for 3D-Aware Image Synthesis," the authors propose a different sampling strategy, which does not require downsampling. Would this strategy perform better? Worse?

3. Section 4.1, Segmentation Experiment Results. It would help to discuss the shortcomings of slot-attention, namely, why the method fails on the segmentation task. My guess why uORF works better would be due to the two disjoint latent spaces for foreground and background? I think this should be highlighted here.

4. Section 4.3, Scene Design and Editing Setup. I did not understand how the setup for modifying the foreground object pose/appearance works. I think you can switch the latent embeddings for background, as you have a one-to-one mapping, but this is not the case for the foreground objects. Could you explain this process in more detail?

5. Appendix B2, Coordinate Space. Here you mention that you use a foreground box to encourage disentanglement of foreground and background slots. How does this foreground box work, what is its influence on the final result? Mentioning this (maybe) crucial detail only in the appendix is not sufficient in my opinion and should be better explained in the main text.


**Summary Of The Paper:**

The paper introduces an interesting new research direction of factorized, 3D consistent neural representations. In particular, it proposes to combine slot-attention mechanisms with conditional neural radiance fields to segment and render novel views of a scene from a single input view. The authors also address one apparent shortcoming in the slot-attention paper: the background and foreground object latent codes are sampled from the same distribution, leading to breakdowns on scenes with complicated backgrounds. The paper proposes to learn two disjoint distributions, one for the background and one for the foreground, to alleviate this issue.

**Summary Of The Review:**

I vote to accept this paper for publication at ICLR 2022. I like the idea of modeling scenes as a combination of disjoint objects, which can be added, removed, modified, and recombined to form new scenes. I also think the paper is well written, well-motivated, and provides extensive experiments. In my opinion, the paper adds to the literature on neural scene representation/decomposition and is interesting to the community. I have some minor suggestions and questions (see above), which I hope the authors clarify during the rebuttal.

---

> ### Author Response · Authors · 2021-11-20
> **Author response to Reviewer pN74**
>
> Thank you for your constructive review and helpful suggestions!
>
> **Q1**: Discussion of Stelzner et al. (2021)
>
> **A1**: Thank you for bringing this up. We totally agree with you on the importance of discussing the concurrent work from Stelzner et al. (2021), and this is why we tried to clarify the relevance and difference in the related work section, despite that it has not been published. We also wish to include a comparsion, but couldn't do so for the exact reason that you mentioned: the code/data for their approach is not publicly available, making comparisons extremely difficult. We appreciate your understanding. Shall the code/data of Stelzner et al. (2021) is made available in the future, we will be more than happy to include such a comparison.
>
> **Q2**: How many samples per ray are used?
>
> **A2**: We used 64 samples per ray. We left this detail in Appendix B.4, and are happy to move it to the main paper.
>
> **Q3**: Do you still use two nets for ray evaluations as NeRF?
>
> **A3**: No, we only use a single network for better efficiency. Our model only requires one forward pass with a single network to render a pixel.
>
> **Q4**: Would GRAF's sampling method work for your method?
>
> **A4**: Thanks for the suggestion! We actually had thought about and tried this idea. It turns out that GRAF's sampling is suboptimal in our setup. Specifically, if we use their sampling with a stride of $1$, then it is effectively the same as our patch-based fine-training; with a stride $>1$, we have observed that their sampling method (as shown in Fig. 3 of the GRAF paper) introduces aliasing in the rendering outputs. This was not an issue in GRAF, because their GAN-style training setup involves training a discriminator together with the renderer. Instead, we are using a perceptual loss that is computed via a network pretrained on ImageNet.  Because this network has not seen such aliasing, the preceptual loss becomes less effective and leads to suboptimal reconstructions.
>
> **Q5**: Is better segmentation results due to two disjoint latent spaces for foreground and background?
>
> **A5**: Yes, this aligns with our ablation studies. From the qualitative comparison in Figure 4, we see that Slot Attention and "ours (w/o background modeling)" manifest similar artifacts. Furthermore, from Table 2, we can see that "ours (w/o background modeling)" performs at a similar level ($11.7\%$ ARI on CLEVR-567) as slot-attention ($3.5\%$ ARI), and "ours" ($83.7\%$) is much better than both. This observation is consistent across all datasets. We have added this analysis to the experiment section in our updated submission (highlighted in blue).
>
> **Q6**: How do you modify foreground object pose/appearance in scene editing?
>
> **A6**: We have shown two types of modification in the submission: changing object position (translation) and adding/removing objects. The change of object position, as shown in Figure 6, is done by imposing a displacement field to the object to be moved. Specifically, if we want to move an object by displacement $[dx,dy,dz]=[1,2,3]$, we inversely displace all query point coordinates to the object radiance field by $[-1,-2,-3]$. Adding/removing objects is done by adding/removing a latent embedding of a foreground object, as shown in Figure 1. It is also possible to rotate an object radiance field. Directly rotating the field will make the object rotate around the scene center in a world coordinate frame. To make the object rotate in the object coordinate frame, we may first compute the baricenter of the density function learned by the radiance field to identify the center of the object, and then apply rotation to be centered around the object itself.
>
> **Q7**: How does foreground box affect final results? Should explain it in main text.
>
> **A7**: Thanks for your suggestion! We set the foreground box in early training to prevent foreground slots from representing the background.  Without the foreground box, the foreground NeRF can attach a piece of background segment to an object slot.  We added a visual comparison to demonstrate this effect in Figure 8 in our updated submission. We have also included the explanation in the main text (highlighted in blue).
>
> Thanks again for your constructive comments. We hope that our responses are helpful in answering your questions. Please don't hesitate to let us know if there are any additional questions.

---

> > ### Comment · Reviewer_pN74 · 2021-11-24
> > **Follow up**
> >
> > Dear authors, reviewers, and area chairs
> >
> > Thank you for your detailed explanations and clarifications. I had a look at the revised version of the paper and the clarifications are addressing my concerns/questions. I wanted to reiterate my belief that this paper should be accepted to ICLR, especially regarding the below borderline score of reviewer one. I do agree with the authors that the comparison with GIRAFFE is unfounded, as the two models address fundamentally different problems (unconditional scene generation vs conditional scene decomposition) and they explicitly address these differences in the appendix. Overall, the paper is a strong submission, providing multiple contributions that I think the community will find insightful.

---

> > > ### Author Response · Authors · 2021-11-28
> > > **RE: Follow up**
> > >
> > > Dear Reviewer pN74,
> > >
> > > Thank you very much for your helpful reviews and your supportive comments! We are happy that our revision has addressed your concerns. We sincerely appreciate your constructive suggestions.
> > >
> > > Thank you!
> > >
> > > Paper396 Authors

---

### Official Review · Reviewer_tkSe · 2021-11-02

**Correctness:** 4
**Technical Novelty And Significance:** 3
**Empirical Novelty And Significance:** 3
**Recommendation:** 8
**Confidence:** 3

**Main Review:**

The authors presented a novel idea. The system is engineered well and the authors have shown success on three synthetic datasets and various applications.
My concerns are as follows.
1. All experiments are conducted on synthetic datasets. Both training and testing use the same set of object shapes - only the arrangements of the objects are different. It is not clear how this approach can generalize to real world scenes. As in the real world, lots of objects have not been seen in the training set. An ablation study that adds unseen shapes into the testing scenes can be very informative. Additionally, a demonstration of the approach on real world scenes would be a strong result to show in the paper (either it is negative or positive).
2. How is the number of the foreground objects decided? Does it have a strong impact in the results?

Edit: The authors addressed my concerns in their revision. In particular, the authors showed additional results on a real world image. As expected, the rendering is not as good as on synthetic data. However, I do not think this overshadows the contribution of this paper - instead, it shows the value as well as limitation of the proposed method, and can inspire future work.

**Summary Of The Paper:**

This paper utilized the powerful NeRF. The authors present a new approach to learn the scene arrangement in an unsupervised way. The training is performed on unlabeled datasets of similar objects in different arrangements. The inference requires only one RGB image as input, and can correctly deduce the arrangement and the 3D geometry of the objects. The authors showed two supporting technical contributions to the system: (1) splitting background and foreground objects leads to better results, (2) a coarse-to-fine training to alleviate the space and time needs. The authors showed success on three synthetic datasets and various applications.

**Summary Of The Review:**

The authors presented a novel idea. The system is engineered well and the authors have shown success on three synthetic datasets and various applications. However, all evaluation and experiments are done on synthetic datasets with the same set of objects and background. Thus it is unclear how this approach can generalize to solve real world problems.

Edit: The authors addressed my concerns in their revision. I think it is a good paper and should be accepted.

---

> ### Author Response · Authors · 2021-11-20
> **Author response to Reviewer tkSe**
>
> Thank you for your constructive review and helpful suggestions!
>
> **Q1**: Testing on unseen shapes would be informative.
>
> **A1**: Thank you for this suggestion! We followed your suggestion and constructed another testset for Room-Diverse. All test objects are drawn from a pool of 500 ShapeNet chairs that are completely disjoint from the training objects. All other settings are the same as the original testset. The results are
>
> -|ARI$\uparrow$ (segmentation metric)|LPIPS$\downarrow$ (novel view synthesis metric)
> :-|:-:|:-:
> Ours, tested on seen shape testset| 65.61 | **0.1729**
> Ours, tested on unseen shape testset| **66.10**| 0.1771
>
> As we can see, results on seeen and unseen shapes are very close, suggesting that our model generalizes well to unseen object shapes. We have added these results to our appendix D and highlighted the new contents in blue.
>
> **Q2**: Demonstration on real world scenes would be strong results.
>
> **A2**: Thank you for this suggestion! We take photos by a cellphone and test our model (trained on Room-Diverse dataset) on real photos. We show results in Appendix D, Figure 10, in our updated submission (New contents are highlighted in blue). As we can see, our model can obtain reasonable results even on real images.
>
> **Q3**: How is the number of foreground objects decided?
>
> **A3**: Based on Slot Attention [Locatello et.al., NeurIPS'20], we actually do not need to decide the number of foreground objects. The model essentially learns a prior distribution; it then samples $K$ objects from this learned distribution as the initialization for the grouping process to allocate them to individual objects. Empty slots are allowed; thus, $K$ can be seen as the maximum number of objects in the scene. Further, this process makes it possible to set a different $K$ during training and testing.
>
> We have demonstrated such generalization through experiments. On CLEVR-567 (each scene has 5--7 objects), we set the number of slots $K$ to 7 throughout training. Then, we tested our model (trained on CLEVR-567 with $K=7$ slots) on scenes with 11 packed objects by setting the number $K$ to 11 during inference. Our model performs reasonably well on such generalization, as shown in Table 5 in main paper and Figure 19 in the updated Appendix (was Figure 17 in the original Appendix).
>
> Thanks again for your constructive suggestions. We hope that our responses are helpful in answering your questions. Please don't hesitate to let us know if there are any additional questions. We also understand that some of the results currently in Appendix may be critical, and are happy to move them to the main paper if needed.
>
> Reference:
>
> *[Locatello et.al., NeurIPS'20] Francesco Locatello, Dirk Weissenborn, Thomas Unterthiner, Aravindh Mahendran, Georg Heigold, Jakob
> Uszkoreit, Alexey Dosovitskiy, and Thomas Kipf. Object-centric learning with slot attention. NeurIPS, 2020*

---

> > ### Comment · Reviewer_tkSe · 2021-11-29
> > **Change my score to accept**
> >
> > Thanks for the rebuttal and in particular the additional results on real world data. They are very clear and addressed all of my concerns. The result on the real world data is not as good as the ones on synthetic data, which is expected. However, I do not think this overshadows the contribution of this paper - instead, it draws the boundary of the contribution and can inspire future work.
> >
> > I think the revised paper is a good paper and can inspire the community if it is published at ICLR 2022. I edited my review and changed the final score to be 8: accept, good paper
> >
> > Thanks,
> > Reviewer tkSe

---

> > > ### Author Response · Authors · 2021-11-29
> > > **Thank you**
> > >
> > > Dear Reviewer tkSe,
> > >
> > > We would like to thank you again for your constructive review. We are happy to see that our revision has addressed your concerns and the real-world results are helpful. We sincerely appreciate your suggestions.
> > >
> > > Thank you!
> > >
> > > Paper396 Authors

---

### Official Review · Reviewer_BYk3 · 2021-11-05

**Correctness:** 3
**Technical Novelty And Significance:** 2
**Empirical Novelty And Significance:** 2
**Recommendation:** 5
**Confidence:** 3

**Main Review:**

strengths:

The overall direction is promising, and factorize the scene representation is indeed an important issue to study.
The technique proposed is sound overall with soft-kmeans like strategy to generate corresponding features in an unsupervised manner, through the GRU probably break the theoretical guarantee of convergence.

Weak:

The overall results looks more in a concept proof, the objects in all test datasets are relatively simple having uniformed color. Feel it can hardly work in real scene senario, as shown in GIRAFF paper.  Under these senarios, the segmented results and rearranged results lose many object details yielding blurry or incorrect renderring.  I feel there should be more improvement over these issues.

The overall concept is fairly close to GIRAFF and major difference could be the training scheme inference from a single image or multiple. I would like to see there could be additional techniqual improvement especially some high resolution representations. Or improvement of architectures in order to support better quality.

some questiions:
Does the algorithm always obtains a reasonable representation w.r.t different initial sampled centers ?





**Summary Of The Paper:**

The paper proposes a novel neural representation of an given image, which decomposes K object instances from background. So various tasks such as rerendering, rearrangement etc.

The learning process is first sampling K centers, and represent each object as a learnable hidden variable z,   an gaussian based soft. k-means styple clustering is then performed afterwards. The differences here are 1)  for sampling the centers, there are  learnt forground and background priors, which can benefit the initial state in this learning process.  2)  updating the centers z with a learnable GRU rathor than simple mean pooling, which I believe it have more flexible representations for the cluster. Finally,  the object clusters are discoverred. In this process.

The author evaluted with 3 self created datasets and show several reasonable results by performing mentioned tasks such as 3D segmentation, rearrangement etc.


**Summary Of The Review:**

This paper points out a good direction to dive into unsupervised learning of compositional scene representations. However, technique strongness and novelty may need to be further improved.

---

> ### Author Response · Authors · 2021-11-20
> **Author response to Reviewer BYk3**
>
> Thank you for your constructive review and helpful suggestions!
>
> **Q1**: Relation to GIRAFFE.
>
> **A1**: We would like to clarify that although our work is related to GIRAFFE, it addresses a **fundamentally different** problem than GIRAFFE, as detailed in **Appendix E**:
> - GIRAFFE targets at unconditional generation; it is not designed for, and therefore **cannot** tackle the problem of inferring object representations from (or conditioned on) either a single image or multiple. In contrast. we focus on such inference from a single image of a multi-object scene.
> - Technically, our model preserves the multi-view consistency of 3D object representations. Our progressive training mechanism further helps to produce output at a higher resolution. In contrast, GIRAFFE's feature field technique trades multi-view consistency for fine details; their generated objects may look differently across different views, and this is why GIRAFFE cannot be used for novel view synthesis (see Figure 25).
>
> In Appendix E, we have also shown comparisons with GIRAFFE to demonstrate these fundamental differences.
> - In Appendix E.1, we show that the single-image inference problem is highly non-trivial and GIRAFFE cannot do it. The observation is that GIRAFFE cannot even do input-view reconstruction, let alone novel view synthesis or segmentation in 3D.
> - In Appendix E.2, we show that GIRAFFE cannot do wide-baseline novel view synthesis even using author-provided pretrained models. This is because their neural renderer sacrifices 3D consistency for higher-resolution images.
> - In Appendix E.3, we analyze why GIRAFFE cannot do inference and novel view synthesis.
>
> If possible, we would sincerely appreciate your time in reviewing Appendix E of our submission. We hope that it will resolve your concern. We also understand that such clarification might be critical, and are happy to move the discussion to the main paper if needed.
>
> **Q2**: Scene complexity and the quality of segmentation and re-arrangement results.
>
> **A2**: Regarding applicability to real scenes, we show a demonstration in Figure 10 in our updated submission. As we can see, our model can obtain reasonable inference results even on real images, yielding reasonable segmentations. Note that GIRAFFE is not able to perform such inference, even on synthetic images (see our answer above and Appendix E). Other previous methods cannot infer unsupervised factorized 3D scene representations from a single synthetic image, either. Thus, we wish to clarify that our proposed method is indeed making progress on a challenging task. We surely agree that the segmentation and re-arrangement quality can always be further improved and we look forward to keeping pushing this frontier.
>
> **Q3**: Is the method robust to different initial centers?
>
> **A3**: Yes! Following your suggestion, we have tested the robustness of our model on the Room-Chair dataset. For each test scene, we now use 5 different random seeds for sampling initial centers. We compute the mean $\mu$ and std $\sigma$ of ARI (segmentation metric) over the 5 seeds. We average them over 500 test scenes. The averaged mean $\bar\mu$ of ARI is $78.2\%$ and $\bar\sigma$ is $1.7\%$. The mean ARI suggests good segmentation results (similar to $78.8\%$ as reported in Table 2 in our main paper), and $\bar\sigma=1.7\%$ indicates that our method consistently achieves the results w.r.t. different initial centers. We have added this analysis to a section in Appendix D in our updated submission (highlighted in blue).
>
> Thanks again for your constructive suggestions. We hope that our responses are helpful in addressing your concerns. Please don't hesitate to let us know if there are any additional questions.

---

> ### Author Response · Authors · 2021-11-24
> **Follow up**
>
> Dear Reviewer BYk3,
>
> Thank you for your time and effort in reading our response! We hope our response has addressed your concerns. If you still feel unclear or concerned, please kindly let us know and we are more than glad to further clarify and discuss any further concerns. If you feel your concerns have been addressed, please kindly consider if it is possible to update your score.
>
> Thank you!
>
> Paper396 Authors

---

> ### Author Response · Authors · 2021-11-28
> **Follow up reminder**
>
> Dear Reviewer BYk3,
>
> We appreciate your time and effort in reading our response and revision! If you still have further concerns or feel unclear, please kindly let us know and we are happy to further clarify and discuss. If you feel your concerns have been addressed, we would appreciate it if you might kindly consider updating the score. As the discussion deadline is in 2 days, we really look forward to your feedback.
>
> Thank you!
>
> Paper396 Authors

---

### Author Response · Authors · 2021-11-21
**Summary of Revisions**

We thank the reviewers again for their helpful suggestions and constructive reviews. We are eager to incorporate their constructive feedback as revisions to our paper (highlighted in blue in the updated submission). We now summarize the revisions in below:

### Experiments

- [BYk3,tkSe] In Appendix D, we demonstrate the inference on a single real photo by our model. The results suggest that our model can reasonably generalize to real images.
- [BYk3] In Appendix D, we analyze the sensitivity to slot initializations. We observe that our model can obtain good results across different slot initializations.
- [tkSe] In Appendix D, we discuss testing results on unseen shapes, indicating that our model generalizes well to unseen object shapes.
- [pN74] In Appendix D, we perform an ablation study on foreground locality box. A visual comparison demonstrates that the foreground box prevents foreground object slots from reconstructing background segments.

### Writing

- [pN74] In the experiment section, we include the discussion that our better segmentation results are due to disjoint background/foreground modeling.
- [pN74] In the method section, we introduce why and how we set the foreground locality box.

In individual responses, we also answered questions from each reviewer and made additional clarifications. We hope that our responses and revisions are helpful. We sincerely appreciate your time and efforts in helping us to improve our paper. Please don't hesitate to let us know if you have any additional questions!

---

### Decision · Program_Chairs · 2022-01-20

**Decision:**

Accept (Poster)

**Comment:**

This paper develops a method for decomposing scenes into object-specific neural radiance fields.  After the discussion phase, two reviewers support acceptance.  Empirical results on multiple synthetic datasets and benchmarks appear convincing; the rebuttal also added an initial demonstration of generalization to real images.